# 9-year trends of PM$_{10}$ sources and oxidative potential in a rural background site in France

Lucille Joanna Borlaza[1*], Samuël Weber[1], Anouk Marsal[1], Gaëlle Uzu[1], Véronique Jacob[1], Jean-Luc Besombes[2], Mélodie Chatain[3], Sébastien Conil[4], and Jean-Luc Jaffrezo[1*]

[1]University Grenoble Alpes, CNRS, IRD, INP-G, IGE (UMR 5001), 38000 Grenoble, France

[2]Université Savoie Mont-Blanc, CNRS, EDYTEM (UMR5204), 73000 Chambéry, France

[3]Atmo Grand Est, 67300 Schiltigheim, France

[4]ANDRA, DRD/GES Observatoire Pérenne de l'Environnement, 55290 Bure, France

*Correspondence to*: Lucille Joanna Borlaza (lucille-joanna.borlaza@univ-grenoble-alpes.fr) and Jean-Luc Jaffrezo (jaffrezo@univ-grenoble-alpes.fr)

**Abstract.** Long-term monitoring at sites with relatively low particulate pollution could provide an opportunity to identify changes in pollutant concentration and potential effects of current air quality policies. In this study, a 9-year sampling of PM$_{10}$ (particles with an aerodynamic diameter below 10 µm) was performed in a rural background site in France (Observatoire Pérenne de l'Environnement or OPE) from February 28, 2012 to December 22, 2020. The Positive Matrix Factorization (PMF) method was used to apportion sources of PM$_{10}$ based on quantified chemical constituents and specific chemical tracers analysed on collected filters. Oxidative potential (OP), an emerging health-metric that measures PM capability to potentially cause anti-oxidant imbalance in the lung, was also measured using two acellular assays: dithiothreitol (DTT) and ascorbic acid (AA). The sources of OP were also estimated using multiple linear regression (MLR) analysis. In terms of mass contribution, the dominant sources are secondary aerosols (nitrate- and sulphate-rich) associated with long-range transport (LRT). However, in terms of OP contributions, the main drivers are traffic, mineral dust, and biomass burning factors. There is also some OP contribution apportioned to the sulphate- and nitrate-rich sources influenced by processes and aging during LRT that could have encouraged mixing with other anthropogenic sources. The study indicates much lower OP values than in urban areas. A substantial decrease (58% reduction from year 2012 to 2020) in the mass contributions from the traffic factor was found, even though this is not clearly reflected in its OP contribution. Nevertheless, the findings in this long-term study in the OPE site could indicate effectiveness of implemented emission control policies, as also seen in other long-term studies conducted in Europe, mainly for urban areas.

## 1 Introduction

Particulate matter (PM) pollution causes various environmental concerns affecting public health and climate. An overwhelming part of the scientific literature on PM chemical characterization and sources focuses with urban and populated

areas, where most emissions originate from and where populations are impacted. Further work has also been carried out in more specific areas to understand particular processes of aerosol formation and transport, as well as specific sources such as in the boreal forest (Yan et al., 2016), polar environments (Barrie and Hoff, 1985; Moroni et al., 2016), high altitude (Rinaldi
et al., 2015), or marine sites (Scerri et al., 2016). Rural sites are of great interest as well because they can represent the regional background of the atmosphere and potential influence from long-range transport (LRT) of pollutants. Studies at such sites provide more understanding of large-scale and mesoscale processes (Anenberg et al., 2010; Mues et al., 2013; Konovalov et al., 2009), which can be useful in chemical transport models.. The continuing can lead to the identification of long-term trends and the effect of recent changes in the source emissions. Indeed, several programmes have been set to monitor atmospheric
composition in a harmonised way for background areas throughout Europe and North America. Among these are ACTRIS (Aerosol, Clouds and Trace Gases Research Infrastructure) (Pappalardo, 2018), EMEP (European Monitoring and Evaluation Programme) (Aas et al., 2012; Alastuey et al., 2016), IMPROVE (Interagency Monitoring of Protected Visual Environments) (Hand et al., 2012), and CAPMoN (Canadian Air and Precipitation Monitoring Network) (Nejedlý et al., 1998). However, only few sites provide long-term in-depth series of PM chemical speciation data.

Further, although current air quality standards are based on ambient mass concentration of PM, there is also a growing interest in new types of characterization that take into account not only particle composition but also its capability to generate health impacts (Park et al., 2018; Crobeddu et al., 2017; Møller et al., 2010; Bates et al., 2019). This is the case of PM oxidative potential (OP) (Nel, 2005; Conte et al., 2017; Yang et al., 2014; Verma et al., 2014), the ability of PM to generate reactive oxygen species (ROS) leading to PM-induced oxidative stress in the lungs. In France, several studies have reported about OP
of ambient PM in different urban environments (Weber et al., 2018, 2021; Borlaza et al., 2021; Calas et al., 2019; Daellenbach et al., 2020), but there are still limited studies performed in rural areas. The characterization of PM sources and OP in a rural site will enable us to investigate the large-scale effects of mitigation policies that target reduction of PM mass concentrations. This will also provide knowledge of the efficiency of current air quality guidelines in terms of other emerging health-based metrics of PM exposure.

The understanding of trends of PM are essential to evaluate the effects of mitigation policies on air pollution levels. A reference background site offers a good opportunity to gauge the broad effects of certain improvements in the transportation fleet and other regulations aimed at reducing vehicular emissions in large cities. Thus, in this study, an extensive dataset of PM over a 9-year period ($n = 434$), obtained from a French national background site, was investigated to: (1) provide insights on the long-term trends of PM sources and other emerging health-based metrics of PM exposure, such as OP of PM, (2) quantify the
temporal evolution of the contributions of these sources, particularly focusing on vehicular emissions that have already been shown to decrease in urban environments in Europe during the last decades.

## 2 Methodology

### 2.1 Site description and sampling parameters

The OPE (Observatoire Pérenne de l'Environnement) sampling site is managed by the French national radioactive waste
management agency (ANDRA). It is located in a remote area in the north-eastern part of France (48.5°N, 5.5°E) at an altitude
of 390 m above sea level, in a large agricultural area without any residential areas within several kilometres (Figure 1). The
mean annual temperature between 2011 and 2018 in the area was 10.5°C [minimum, maximum: -15.2°C, 36.4°C], average
cumulated yearly precipitation was 829 mm, and the predominant local wind regimes are south-westerly and east-north-
easterly winds (Conil et al., 2019). This site, being far from any local anthropogenic sources, is considered representative of
the rural atmospheric background of north-eastern France. Golly et al. (2019) also demonstrated that the PM chemistry at this
site is very close to that in several other background rural sites in France.

The $PM_{10}$ samples in this study were collected from February 28, 2012 to December 22, 2020. Particularly, from February 28,
2012 to December 28, 2015, the samples ($n = 181$) were collected on a weekly basis (from Tuesday 9:00 AM to Tuesday 9:00
AM) using a low volume sampler (Partisol, 1 $m^3 h^{-1}$) onto 47 mm-diameter quartz fibre filters (Tissuquartz PALL QAT-UP
2500 diameter 47 mm). From January 12, 2016 to December 22, 2020, the samples ($n = 253$) were collected on a daily (24-
hour) basis in a 6-day sampling interval using a high-volume sampler (Digitel DA80, 30 $m^3 h^{-1}$) onto 150 mm-diameter quartz
fibre filters (Tissuquartz PALL QAT-UP 2500 diameter 150 mm).

All filters were preheated at 500 °C for 12 hours before use to avoid organic contamination. Blank filters (about 10% by
number of the actual filters) were also collected to quantify detection limits and to secure the absence of contamination during
sample transport, setup, and recovery. After collection, all filter samples were wrapped in aluminium foil, sealed in zipper
plastic bags, and stored at <4 °C until further chemical analysis.

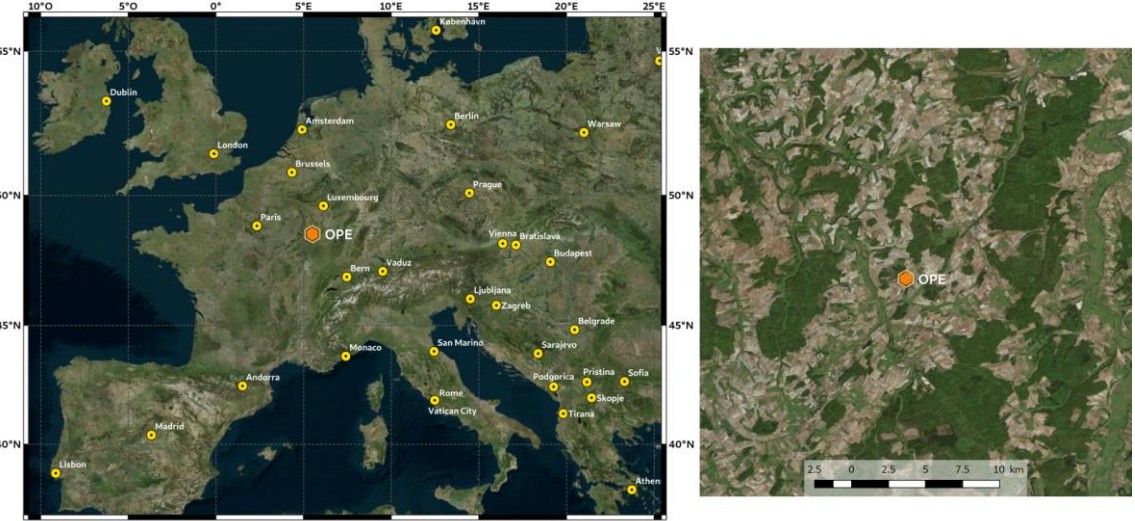

**Figure 1: Location of the OPE site in France. ©OpenStreetMap contributors 2020. Distributed under a Creative Commons BY-SA License**

## 2.2 Chemical analyses

After collection, samples were subjected to various chemical analyses to perform the quantification of the major constituents by mass and specific chemical tracers of sources needed for the PMF model. These analyses were performed in the same laboratory for all samples ($n$=434) during the entire sampling duration (February 28, 2012 to December 22, 2020).

The carbonaceous components (organic carbon (OC) and elemental carbon (EC)) were analysed using a thermo-optical method on a Sunset Lab analyser (Birch and Cary, 1996), using the EUSAAR2 temperature program. Total organic matter (OM) in daily ambient $PM_{10}$ were estimated by multiplying the OC mass concentration by a factor 1.8. Yazdani et al. (2021) showed that this is consistent with the range estimated for rural samples from the IMPROVE network, that are generally higher than for urban samples.

A set of other chemical analyses was performed on a single water extraction of each filter. This extraction was performed in a 10 ml of ultra-pure water under vortex agitation for 20 minutes. The extract was then filtered with a 0.22 µm porosity Nucleopore filter. The major ionic components ($Na^+$, $NH_4^+$, $K^+$, $Mg^{2+}$, $Ca^{2+}$, $Cl^-$, $NO_3^-$, $SO_4^{2-}$) and methane sulfonic acid (MSA) were measured by ion chromatography (IC, Thermo Fisher ICS 3000) following a standard protocol described in Jaffrezo, et al. (1998) and Waked, et al. (2014). An ICS300 (Thermo-Fisher) with AS11 HC column for the anions and CS16 for the cations was used.

The analyses of anhydro-sugars and primary saccharides were achieved using a high-performance liquid chromatography with pulsed amperometric detection (HPLC-PAD). The samples collected before year 2017 was analysed using a set of Metrohm columns (MetroSep A Supp 15 and Metrosep Carb1) on Dionex DX500 HPLC. The samples collected after year 2017 were analysed with a Thermo-Fisher ICS 5000+ HPLC equipped with 4 mm diameter Metrosep Carb 2×150 mm column and 50 mm pre-column. The analytical run is isocratic with 15% of an eluent of sodium hydroxide (200 mM) and sodium acetate (4 mM) and 85% water, at 1 ml min$^{-1}$. These methods allowed the quantification of anhydrous saccharides (levoglucosan and mannosan) and polyols (sum of arabitol and mannitol) as described in detail in Waked et al. (2014) and Samaké et al. (2019).

Trace elements were analysed after mineralization, using 5 ml of $HNO_3$ (70 %) and 1.25 ml of $H_2O_2$ during 30 minutes at 180 °C in a microwave oven (microwave MARS 6, CEM). The elemental analysis (Al, As, Ba, Ca, Cd, Ce, Co, Cr, Cs, Cu, Fe, K, La, Li, Mg, Mn, Mo, Ni, Pb, Pd, Pt, Rb, Sb, Se, Sn, Sr, Ti, Tl, V, Zn, Zr) was performed on this extract using inductively coupled plasma mass spectroscopy (ICP-MS) (ELAN 6100 DRC II PerkinElmer or NEXION PerkinElmer) as described by Alleman et al. (2010).

All procedures have been performed following the related EN standards (i.e., EN 12341, EN 14902, EN 16909, EN 16913). A quality control of the chemical analyses including a mass closure test, available in the supplementary information (SI, see S1). A summary of the quantification limits (QL) on each chemical specie measured in the OPE site is also provided in Table S1.

Finally, our group successfully and regularly participates in inter-laboratory comparison exercises for OC and EC within ACTRIS, and in EMEP (European Monitoring and Evaluation Programme) for ions analysis. The $PM_{10}$ measurements from the tapered element oscillating microbalance (TEOM) are all in a daily (24-hour, 09:00 to 09:00) resolution, while the

reconstructed PM10 were obtained from chemical analysis performed on filters collected on a weekly (7 days, 09:00 to 09:00) or daily (24-hour, 09:00 to 09:00) basis. A total of 299 out of 434 (69%) TEOM measurements were paired with reconstructed

PM10 data, due to many interruptions in the TEOM functioning, in order to evaluate the semi volatile mass missing in the mass reconstruction with filter chemistry.

## 2.3 Oxidative potential (OP) analysis

The OP analysis was performed on $PM_{10}$ extracts from collected filter samples using a simulated lung fluid (SLF) solution composed of a Gamble + DPPC (dipalmitoylphosphatidylcholine) at 25 µg ml$^{-1}$ iso-mass concentration (Calas et al., 2017).

This methodology facilitates particle extraction in conditions closer to lung physiology. The OP analysis only started on samples collected from June 13, 2017 to December 22, 2020, amounting to a total of 191 samples.

Two assays were used to characterize OP activity: (1) dithiothreitol (DTT) and (2) ascorbic acid (AA) assays, as briefly described in the following sections. The volume-normalized OP activity ($OP_v$) is the OP consumption (nmol min$^{-1}$) normalized by the sampled air volume (m$^3$), representing the OP exposure in each sample. All samples analysed were subjected to triplicate

analysis and each sample results in the mean of such triplicate. The common coefficient of variation (CV) is between 0 and 10% for each assay.

DTT is used as a chemical surrogate to mimic *in vivo* interaction of PM with biological reducing agents, such as adenine dinucleotide (NADH) and nicotinamide adenine dinucleotide phosphate (NADPH), in the DTT assay. The consumption of DTT in the assay represents the ability of PM to generate ROS (i.e., superoxide radical formation) (Cho et al., 2005). The $PM_{10}$

extract is mixed with the DTT solution. Afterwards, the remaining DTT that did not react with $PM_{10}$ is reacted with 5,50-dithiobis-(2-nitrobenzoic acid) (DTNB). This reaction produces 5-mercapto-2-nitrobenzoic acid or TNB. The TNB is measured by absorbance at 412 nm wavelength using a plate-reader (TECAN spectrophotometer Infinite M200 Pro) with 96-well plates (CELLSTAR, Greiner-Bio) in a 10-minute time step interval for a total of 30 minutes of analysis time.

AA is a known antioxidant used in AA assays using a respiratory tract lining fluid (RTFL) (Kelly and Mudway, 2003). This

antioxidant prevents the oxidation of lipids and proteins in the lung lining fluid (Valko et al., 2005). The consumption of AA also represents PM-induced depletion of a chemical proxy (i.e. cellular AA antioxidant). The mixture ($PM_{10}$ extracts reacted with AA) is injected into a 96-well multiwall plate UV-transparent (CELLSTAR, Greiner-Bio) and measured at 265 nm absorbance using a plate-reader (TECAN spectrophotometer Infinite M200 Pro) in a 4-minute time step interval for a total of 30 minutes of analysis time.

Both DTT and AA assays measure OP by depletion of specific chemical proxies, cellular reductants (for DTT) and antioxidants (for AA). Studies have well-identified a large number of PM constituents that influence OP concentrations. At least, OP assays are known to be associated with some metals (Cu, Fe, Mn among others) and some organic species (especially photochemically sensitive species such as quinones) (Calas et al., 2017, 2019, Charrier et al., 2014, Pietrogrande et al., 2019). However, in ambient air, each assay reports its own associations that may vary according to the local context (emission sources, local

transport leading to various ageing processes and spatiotemporal variations) (Gao et al., 2020). Hence, a synergetic approach

using multiple OP assays, to capture the most complete information regarding PM reactivity, is commonly suggested. (Bates et al., 2019, Calas et al., 2017, Borlaza et al., 2021)).

In every experiment, a positive control test is performed to ensure the accuracy and precision of measurements. A 1,4-naphthoquinone (1,4-NQ) solution was used for both DTT (40 µl of 24.7 µM stock solution) and AA (80 µl of 24.7 µM 1,4-NQ solution) assays. The CV of the positive controls were <3% for the 2 assays. Additionally, an ambient filter collected from the lab roof (with an expected constant OP value) was added on each microplate to ensure precision of OP measurements.

## 2.4 Source Apportionment

### 2.4.1 PMF model and input variables

The United States Environmental Protection Agency Positive Matrix Factorization (EPA PMF 5.0) software (Norris et al., 2014) was used to identify and quantify the major sources of $PM_{10}$. PMF is a receptor model fully described by Paatero and Tapper (1994) and is now widely used for source apportionment around the world. Additional information about the model description is provided in the SI (S2).

In this study, 23 chemical species were used as input variables, namely OC, EC, ions ($Na^+$, $NH_4^+$, $Mg^{2+}$, $Ca^{2+}$, $NO_3^-$, $SO_4^{2-}$), trace metals (Al, Cu, Fe, Rb, Sb, Se, Sn, Ti, Zn) and organic markers (MSA, levoglucosan, polyols (sum of arabitol and mannitol)). We assumed that arabitol and mannitol came from a similar source and, hence, combined them into one component named as "polyols" (Samaké et al., 2019). The uncertainties of the input variables were calculated based on Gianini et al. (2012). Finally, the species displaying a signal-to-noise ratio (S/N) lower than 0.2 were discarded and those with S/N between 0.2 and 2 were classified as "weak", consequently multiplying the uncertainties by a factor 3.

### 2.4.2 Criteria for a valid solution

Solutions with a total number of factors from 6 to 11 were tested for the baseline models. Following the recommendations of the European guide on air pollution source apportionment with receptor models (Belis, 2019), the $Q/Q_{exp}$ ratio (<1.5), the geochemical interpretation of the factors, the weighted residual distribution, and the total reconstructed mass were evaluated during factor selection.

Moreover, the bootstrapping method (BS) was used on the final solution to estimate errors and ensure the stability and accuracy of the solutions. The BS method was applied with 100 iterations of the model and contribution uncertainties are presented in the SI (S3) as mean±std of the 100 BS runs. The contribution uncertainties were estimated based on the method presented in Weber et al. (2019) and presented in Figures S2 to S10. The daily specie contributions are estimated using:

$$X_{BSi} = G_{ref} \times F_{BSi}$$

where $F_{BSi}$ is the profile of the bootstrap i, and $X_{BSi}$ is the time series of each species according the reference contribution $G_{ref}$ and the bootstrap run $F_{BSi}$.

Finally, the factor chemical profiles obtained during this study were compared with those from previous studies in France, using the PD-SID method (Belis, 2019; Weber et al., 2019), in order to validate their proper similarity.

### 2.4.3 Appropriate constraints in the PMF model

A set of constraints were applied on a basic model solution, in order to refine the results of the mathematical model by providing sound geochemical knowledge. Hence, the usual constraints as discussed in Weber et al. (2019) and some constraints corresponding to the traffic source following Charron et al. (2019) were also tested on the model (Table S3). However, these set of constraints were tested with caution as most of them have been previously applied only on sites with different typologies (i.e., urban or roadside sites), questioning their applicability in a rural site such as the OPE site. Finally, only a limited set was applied to generate the final solution (S1). After application of the constraints, a BS method was re-applied to verify the stability of the model.

### 2.4.4 Similarity assessment of chemical profiles

To investigate further any differences in the chemical profiles at the OPE site compared to those obtained at other French sites, a test of similarity was performed using the Pearson distance (PD) and standardized identity distance (SID) metric. This is calculated using Eq. S5 and Eq. S6 in the SI (S2) (Belis et al., 2015), closely following a previous work by our group (Weber et al., 2019). This comparison is based on the source relative mass composition, which allows the evaluation of the variability of PMF solutions across different sites. In this case, the chemical profiles obtained for the OPE site were compared against 15 different other sites over France. A "homogenous source" tends to have a similar profile over different site types and should have PD<0.4 and SID<1.0 (Pernigotti and Belis, 2018). Conversely, the sources with PD and SD values outside of this range are considered as "heterogeneous sources".

### 2.5 OP source contribution estimation

The OP contribution of each $PM_{10}$ source was determined by performing an OP deconvolution method using multiple linear regression (MLR) analysis. This methodology is based on the procedure proposed in Weber et al. (2018).. Briefly, the OP activity in nmol/min/m³ was used as the dependent variable, while the PMF-resolved source $PM_{10}$ mass contributions in µg/m³ are the independent variables, as shown in Eq. 1:

$$OP_{obs} = (G_n \times \beta_n) + \varepsilon, \qquad\qquad\qquad\qquad (Eq.\ 1)$$

where $OP_{obs}$ is the observed daily $OP_v$ ($nmol_{reactant}$ $min^{-1}$ $m^{-3}$) with matrix size d×1, G is the PMF-resolved source contribution ($µg$ $m^{-3}$) of size d×n, and β is the regression coefficient representative of the intrinsic OP ($OP_m$) ($nmol$ $min^{-1}$ $µg^{-1}$) of each n source. Finally, $\varepsilon$ is the residual between the observed and modeled OP ($nmol_{reactant}$ $min^{-1}$ $m^{-3}$). The source-specific OP contribution is calculated by multiplying the regression coefficient of each source by the mass contribution of the source to $PM_{10}$ ($G_k \times \beta_k$). This methodology is essentially based on that in Weber et al. (2018).

**2.6 Season-trend (STL) deconvolution method**

The STL (Season-trend deconvolution using locally estimated scatterplot smoothing) model is a versatile and robust statistical method allowing the decomposition of a time-series dataset into three components including trend, seasonality, and residual. The trend provides a general direction of the over-all data; the seasonality is a repeating pattern that recur over a fixed period of time; finally, residual is the random fluctuation or unpredictable change in the dataset. The seasonal component allows to eliminate seasonal variation from the time series, resulting to a smoothed trend line that shows the tendency of the time-series dataset. This method somehow takes into account the changes in seasonal cycles from year to year which could also delineate part of the effect of meteorology on the long-term trend of $PM_{10}$.

To investigate the long-term trends of sources or species concentrations, the STL model (Cleveland et al., 1990) was applied on the monthly mean concentrations, as described by Eq. 2:

$$Y(t) = T(t) + S(t) + r(t) \qquad\qquad\qquad (Eq.\ 2)$$

where : $Y(t)$ is the time series observed in monthly average, $T(t)$ is the trend component of the signal, $S(t)$ is the seasonal component, and $r(t)$ is the residual part not explained by the trend and seasonal part. The frequency was set to 13 (i.e., 6 months before and after the current month) to account for yearly seasonality. This model uses an iterative algorithm that constantly minimizes the residual $r(t)$ by successively adjusting the trend and seasonal components. It has to be noted that the resulting $T(t)$ and $S(t)$ do not represent concentrations, but a statistical deconvolution of the input signal $Y(t)$. $S(t)$ could then be negative and the trend $T(t)$ should be interpreted as an elaborated "moving average" of the concentrations. To account for extreme events or outliers in the data, the impact of data points with very high residuals were given less weight in the estimation of the trend and seasonal components, using the "robust" option of the algorithm. The presence and strength of tendency was evaluated thanks to the ordinary least square (OLS) linear fit of the $T(t)$ component against time, removing the first and last 6 month of the time-series to avoid edge-effects. Note that due to the lack of $PM_{10}$ measurements in July 2019, the concentrations for that month were arbitrarily set to the August 2019 values. This model was implemented in Python 3.8 making use of the *statsmodels* module (Seabold and Perktold, 2010).

**3 Results and discussion**

The sections 3.1 to 3.5 below discuss the concentrations, sources, and trends of $PM_{10}$, while the sections 3.6 and 3.7 discuss the OP measurements and sources.

**3.1 $PM_{10}$ and its major chemical components**

The reconstructed mass of $PM_{10}$ in the OPE site was calculated following Eq. S1 in the SI and is presented in Figure 2. The mass concentration of the reconstructed daily $PM_{10}$ samples ranged from 2 to 51 µg m$^{-3}$ with an overall average of $9 \pm 7$ µg m$^{-3}$ (median: 8 µg m$^{-3}$). These reconstructed $PM_{10}$ mass concentration only exceeded the $PM_{10}$ European limit value of 40 µg m$^{-}$

[3] a few times (*n* = 3) in the entire measurement period. These values are in the lower range of the concentrations reported for rural areas in Europe, ranging from 3 to 35 μg m$^{-3}$ (Putaud et al., 2004), and are relatively close to the values found at a remote site in Revin (France, located 165 km away), as described in the SOURCES programme (average of 13 μg m$^{-3}$) (Weber et al., 2019). Some changes in the concentration can be observed in the PM$_{10}$ mass concentration, but there are no drastic changes in

245    the major chemical components at the OPE, even with the lockdown restrictions during year 2020. The yearly average volatile mass (i.e., unaccounted by chemical analysis), deduced from the difference between TEOM-FDMS measurements and reconstructed PM$_{10}$, ranges from 9% to 44% with an average of 22% (of the yearly median) and is well within range generally found in a rural environment (Pey et al., 2009).

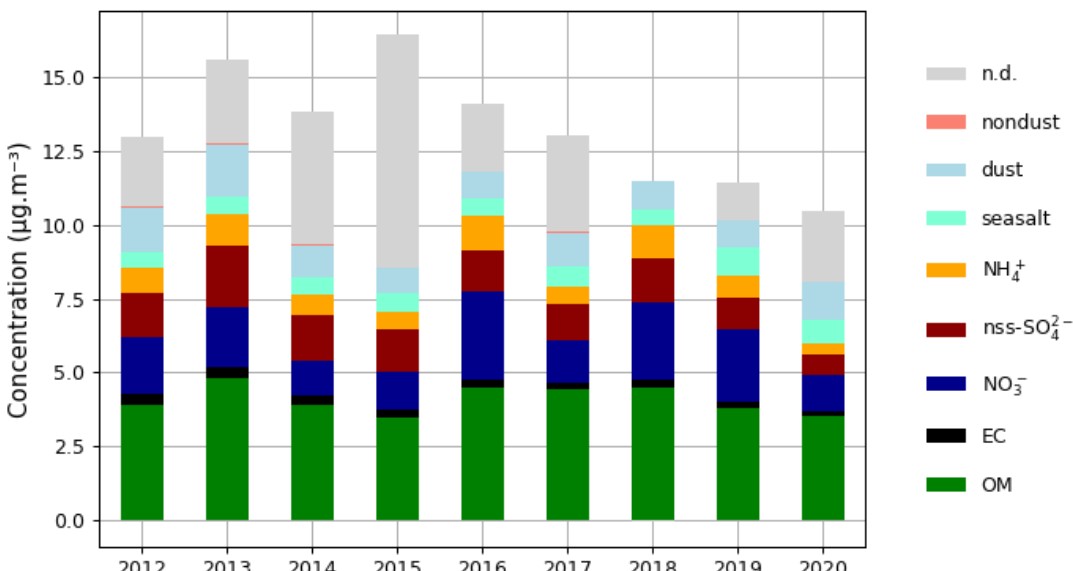

250    **Figure 2: The annual average of PM$_{10}$ composition in the OPE site.**

Accounting for 37% to 45% (based on year) of the reconstructed PM$_{10}$ mass concentrations, organic matter (OM) is the largest contributor. The other main contributors are inorganic secondary species (NO$_3^-$, NH$_4^+$, nss-sulphate), suggesting a strong influence from LRT of pollutants. There are also contributions coming from dust and sea salt. Although all of these components are often dominated by specific emissions, they can be derived from a wide range of sources. For example, vehicular emissions

255    are usually composed of both carbonaceous and metals species, while road dust are usually minerals and some metal species. Understanding the sources (as with the PMF methodology) and transformation processes of PM proves to be an essential step for efficient air quality policies.

### 3.2 Statistical stability of the PMF solution

The final retained solution includes 9 factors as described in section 3.3. Only 71 out of 100 baseline runs (without constraints)

260    converged for this solution, but most factors were 100% correctly mapped, except for the traffic factor (93%, 66 out of 71

converged solutions) and sulphate-rich factor (99%, 70 out of 71 converged solutions). Applying the constraints greatly improved the BS mapping to 100% on all factors. Adding constraints in the base model allowed refining of the model through addition of expert knowledge on the profiles, which lead to the increased model stability. In previous source apportionment studies, specifically by our group, there are common constraints used depending on the site type such as presented in Borlaza et al. (2020).

Particularly, the constraints for traffic-related factors reported in Charron et al. (2019) have been optimized for traffic and urban background sites in our previous works. However, these constraints appeared restrictive when applied in a rural typology such as the OPE site. In fact, the Cu-to-Sb ratios appeared unsuitable as this ratio was 6.3 in our final solution compared to 12.6 in Charron et al. (2019). Based on literature, the Cu-to-Sb ratio can range from 1.6 (Handler et al., 2008) to 12.6 (Charron et al., 2019) depending on site typology. The addition of this constraint resulted in or led to a non-convergent solution in the OPE site. To avoid inconsistencies, the Cu/Sb constraint was excluded in the optimal solution. The OC-to-EC ratio in the traffic profile was also too restrictive for the model, as this ratio was 3.9 in our baseline solution against 0.44 in Charron et al. (2019). The OC/EC levels calculated in this profile also suggests a strong influence of secondary organic aerosols (SOA) (Johnson et al., 2006; Pio et al., 2011; Rodríguez González et al., 2003; Viana et al., 2006), instead of primary traffic emissions. As OC in a rural site can undergo multiple re-transformations in the atmosphere from the emissions sources, this has led to a wide range of OC-to-EC ratios as similarly found in Weber et al. (2019), hence this constraint was excluded.

In the final model, some constraints were used as summarized in Table S3, which resulted to all factors being correctly mapped and all BS runs converged, suggesting a good improvement in the traffic (from 93% to 100%) and sulphate-rich (from 99% to 100%) factors as well as the overall statistical robustness of the model. The other constraints either resulted in a non-convergent constrained model and/or less robust BS results. This implies that in sites with strong influence of LRT, the appropriate constraints tend to vary and an optimal PMF solution can be more difficult to achieve.

The challenge in adding the constraints may also be linked to the inherent nature of the PMF algorithm since it assumes chemical profiles are identical for the whole period of analysis. However, during the 9 years of this study, some chemical source profiles may have changed, notably the traffic factor. Indeed, an evolution of the car fleet in France and Europe could lead to the changes in the OC-to-EC ratio emitted by the vehicle, so this profile may also have changed during this period. For this specific case, a rolling PMF approach (Canonaco et al., 2021) with statistically-mapped PMF profile could be useful to investigate the time-variability of a given profile, slightly evolving with time.

### 3.3 PMF solution description and $PM_{10}$ contributions

The 9 resolved sources of $PM_{10}$ in the OPE site include nitrate-rich (25% of average contribution to $PM_{10}$ for the full period), sulphate-rich (15%), traffic (12%), mineral dust (16%), biomass burning (9%), fresh sea salt (4%), aged sea salt (6%), primary biogenic (7%), and MSA-rich (7%). These factors were identified based on their chemical profiles and the mass loading of specific tracers, as summarised in Table S2 in the SI. The error estimations, chemical profiles, and temporal evolutions of the PMF-resolved sources are available in the SI (S3). Figure 3 represents the repartition of the chemical species in the different

factors. The summed PM$_{10}$ contributions from all sources showed very good mass closure ($r$=0.95) with PM$_{10}$ mass reconstructed with Eq. S1, indicating very good model results.

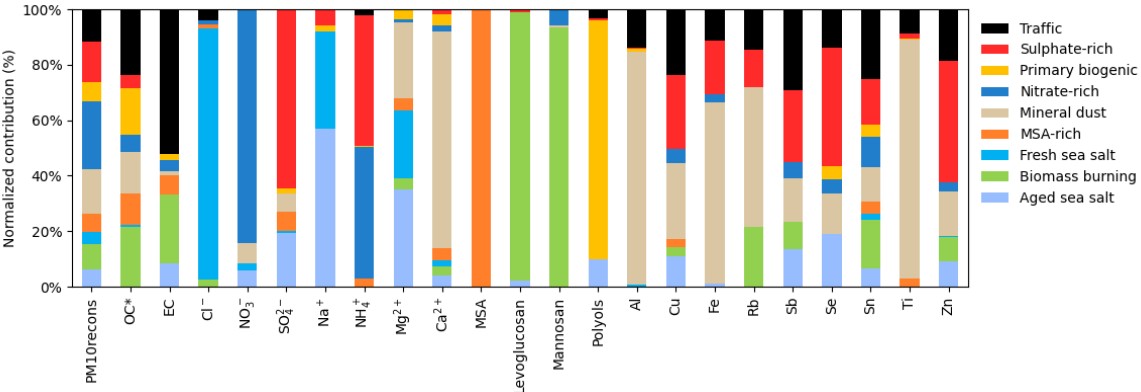

**Figure 3: Species repartition by PMF-resolved sources in the OPE site.**

The factors with highest average contribution to the PM$_{10}$ mass are the two inorganic secondary aerosol sources, nitrate-rich (25%, 2.3±4.3 µg m$^{-3}$) and sulphate-rich (15%, 1.4±1.5 µg m$^{-3}$), and mineral dust (16%, 1.6±1.7 µg m$^{-3}$). Sulphates and nitrates are mainly formed through secondary processes in the atmosphere with long atmospheric lifetimes and can, therefore, originate from regional sources or LRT. Considering the agriculture and natural emissions of ammonia, especially expected in a rural site, secondary aerosols could also be formed locally in the OPE site. The less dominant sources are traffic, biomass burning, biogenic (MSA-rich, primary biogenic), and sea salts (fresh and aged). The contributions of the different factors are quite similar to those observed at other rural sites in France (Weber et al., 2019).

The OPE site has a Northern hemisphere mid-latitude climate with four seasons, (1) winter season corresponding to the months of December, January, February; (2) spring season corresponding to March, April, May; (3) summer season corresponding to June, July and August; and (4) fall season corresponding to September, October and November. Seasonality in some factors can be apparent, such as for the biomass burning and nitrate-rich factors, which are more prominent in winter and spring, respectively, and the primary biogenic and MSA-rich factors, increasing in summer due to greater photochemical and biological activities. Figure 4 depicts the seasonal average contributions of the PM$_{10}$ sources at the OPE site from years 2012 to 2020.

The **nitrate-rich** factor, identified by high loadings of NO$_3^-$ and NH$_4^+$, has a strong seasonal pattern with maximum contribution to PM$_{10}$ mass concentration especially in the months of March and April.

The **sulphate-rich** factor is identified by high loadings of SO$_4^{2-}$ and NH$_4^+$. There are also contributions from some metal species (Se, Zn, Cu, and Sb) in this factor, suggesting potential influence from road dust and/or non-tail pipe vehicular emissions. A small portion of OC (5% of OC mass) is also observed in this factor. The presence of these metals remained, even when the number of factors was increased (up to 11 factors) during the PMF optimization process.

The **aged sea salt** factor is characterised by high loadings of $Na^+$ and $Mg^{2+}$, with a certain amount of species originating from potentially anthropogenic sources such as nitrates (6% of $NO_3^-$ mass) and sulphates (19% of $SO_4^{2-}$ mass) that can be attributed

to mixing and transformation processes in the atmosphere. Interestingly, there are some contributions from EC (8% of EC mass), Cu (11% of Cu mass), Sb (13% of Sb mass), and Se (19% of Se mass). This could imply potential mixing of aged sea salt with other anthropogenic source linked to these species (e.g., traffic, shipping). The minimal loadings observed in the contributions of $Cl^-$ in this factor can be a likely result of ageing processes occurring between sea salt and acidic particulate compounds such as nitric and sulfuric acid (Seinfeld and Pandis, 2016). This factor could also be associated to road salting in

the winter, however there is no clear seasonality in the contributions to support this hypothesis. There was no added constraint in this factor as our solution shows a $Mg^{2+}$ to $Na^+$ ratio at 0.06 while this ratio is usually found around 0.12 in sea-salt emissions (Henderson and Henderson, 2010).

The **fresh sea salt** factor is characterised by high loadings of $Cl^-$ (91% of $Cl^-$ mass) and some contributions from $Na^+$ (35% of $Na^+$ mass) and $Mg^{2+}$ (25% of $Mg^{2+}$ mass). This factor contributes 4% to total $PM_{10}$ mass and, unlike the aged sea salt factor, it

is less likely influenced by anthropogenic sources with extremely low contributions from carbonaceous and metal species.

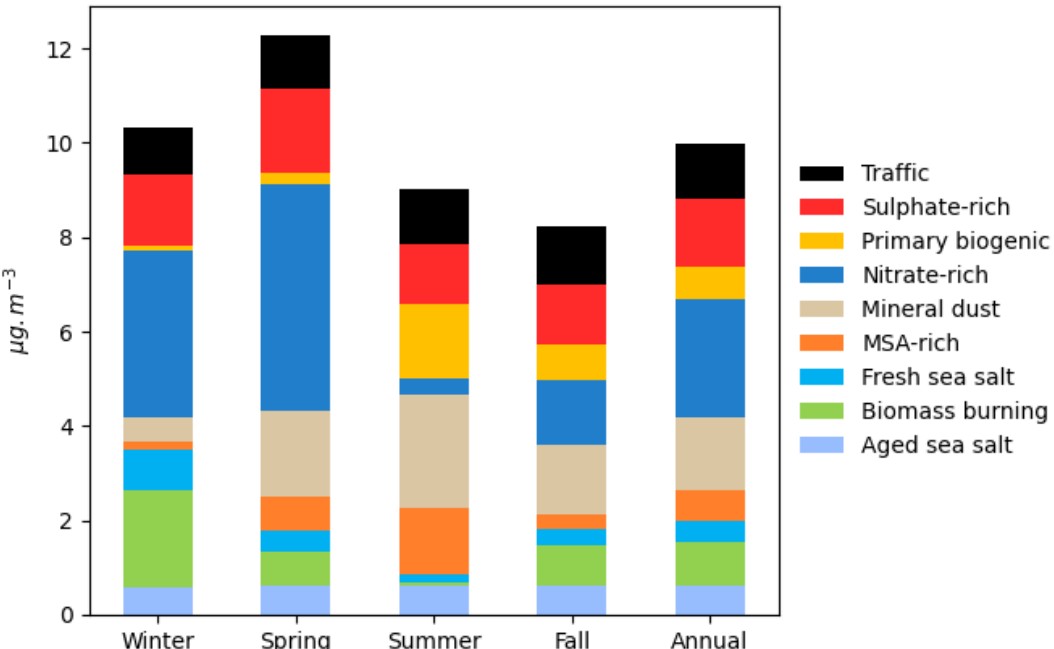

**Figure 4: Seasonal and annual contribution of the PMF-resolved sources in the OPE site.**

The **MSA-rich** factor identified by high loadings of MSA (methanesulfonic acid), a known product of oxidation of dimethylsulfide (DMS) commonly from marine phytoplankton emissions (Chen et al., 2018; Li et al., 1993). A small mass

fraction of $SO_4^{2-}$ (7% of $SO_4^{2-}$ mass) is also found in this factor, that may be due to the co-emission of DMS and non-sea salt sulphates, but also resulting from the production of biogenic sulphate from DMS oxidation. Hence, MSA-rich could potentially be mixed with secondary inorganic aerosols as well. The measured MSA mass concentration showed weak correlations with

specific ionic species from marine aerosols such as $Na^+$ ($r<0.01$) and $Mg^{2+}$ ($r<0.01$). This could indicate that marine biogenic emissions may not be the only source of this factor. Instead, this factor could be influenced by sources with terrestrial origins

and/or from forest biota, as previously reported in other studies (Bozzetti et al., 2017; Golly et al., 2019; Jardine et al., 2015; Miyazaki et al., 2012). This factor also presents a clear seasonal pattern with maximum contribution from May to July due to higher photochemical activity and algae / microbial activity. Golly et al. (2019) reported a very coherent seasonal cycle for MSA concentrations over a large portion of the French territory, including at the OPE site.

The **primary biogenic** factor is characterised entirely by polyols. These species are emitted by fungal spores which partly

explains the high loadings of OC found in this factor. This factor has a higher contribution to $PM_{10}$ during the summer season, consistent with the observations at other rural and urban sites (Samaké et al., 2019; Weber et al., 2019; Borlaza et al., 2021). More details about the characteristics of primary biogenic aerosols can be found in Samaké et al. (2019). Briefly, meteorological conditions, such as high temperature and relative humidity, could facilitate the increase in their formation. This factor can also include some fraction of plant debris, identified by cellulose measurements, as discussed in Samaké et al.

(2019), Borlaza et al. (2021), and Brighty et al. (2021).

The **biomass burning** factor, a major contributor to $PM_{10}$ during the winter season, includes most of levoglucosan and mannosan. This factor contains around 25% of the total EC mass, consistent with a combustion chemical profile. Trace elements like Rb and Sn are also found in this factor, rubidium being the major trace element with 21% of its mass being reconstructed in this factor. Due to the distance of any residential areas from the OPE site, the contributions of this factor to

$PM_{10}$ at 20% on winter average is much less than the contributions generally observed in most sites in France in winter (e.g., urban, sub-urban or countryside sites), which are mainly in the range of 25 - 70 % with an average of about 35% (Weber et al., 2019). The contribution at the OPE site most probably represents the average winter loading of the French national background of the atmosphere.

The **traffic** factor is the second factor where EC is a major contributor (52% of EC mass). The major trace elements found in

the traffic factor are Cu, Sb, Sn, and Zn, and most of their masses are reconstructed in this factor. There is no seasonality associated with this factor. However, its contribution to $PM_{10}$ presents a decreasing trend over the sampling period from 2012 to 2020. This is also consistent with the decreasing trend found in EC mass concentrations over the same period, as presented in Figure S11. These findings present an opportunity to explore the potential decrease in traffic emissions observed in the OPE site, taken as a good proxy of the national background burden of the rural atmosphere. This is further discussed in section 3.5.

The **mineral dust** factor is mainly composed of $Ca^{2+}$ (78% of $Ca^{2+}$ mass), Al (84% of Al mass), and Ti (86% of Ti mass). There are also contributions from other trace elements that could be originating from re-suspended road dust or non-tail pipe emissions such as Fe (65% of Fe mass), Cu (27% of Cu mass), Rb (51% of Rb mass), and Zn (16% of Zn mass). A fraction of it could therefore be of anthropogenic origin.

**3.4 Comparison of the source chemical profiles with previous results in France**

Figure 5 presents the similarity plot (PD-SID distances) obtained for the 9 factors found in the OPE site compared to the French sites included in the SOURCES programme (Weber et al., 2019). It is striking that most factors remained homogeneous within France, including both rural and urban sites. The most stable factors are nitrate-rich (lowest PD) and mineral dust (lowest SID). The traffic factor also appears relatively stable, but presents some dissimilarities according to the high variation in the PD metric. A high PD variation generally indicates a difference in the chemical species that identify the main mass contribution

of the profile. In fact, in the traffic factor, this variation between the OPE and other sites can be attributed to the variations in the OC to EC levels, similar to the findings in Weber et al. (2019). Compared to the other French sites, the OC to EC levels of this factor in the OPE site are much higher, which highlights a strong influence from LRT processes with SOA formation.

The aged sea salt and MSA-rich factors are the only ones positioned outside of the shaded box in Figure 5, indicative of heterogeneous profiles between the OPE and the other sites. The heterogeneity found in the aged sea salt profile can be

attributed to the contributions of EC and some metals in this factor. These were not typically found in other sites in France and could also be due to the mixing of this sea salt profile with other anthropogenic contributions as a result of LRT processes, as well as different aging processes. Similarly, the MSA-rich factor has previously shown site-to-site variations and a wide variation in the PD-SID metric (Weber et al., 2019), mostly attributed to the variability of the contribution of OC in some sites. This is also the case in the OPE site, with a large contribution of about 11% of OC in this MSA-rich factor. Despite these few

differences, the very large similarity of the chemical profiles at OPE compared to those at all other sites in France will be essential to the comparison of the intrinsic OP of sources in section 3.7.

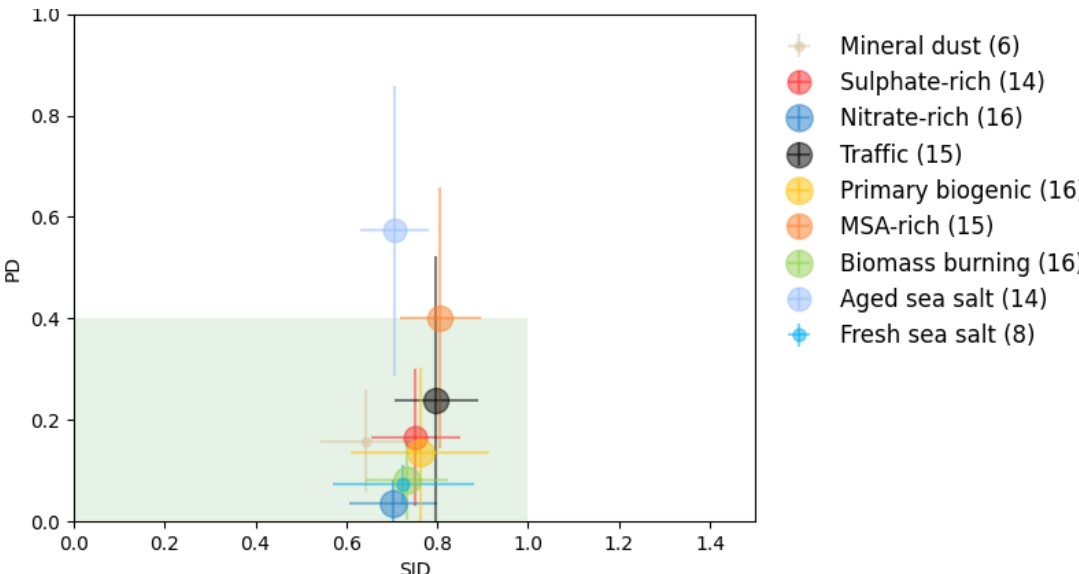

**Figure 5: Similarity plot of the OPE site against all the French sites in the SOURCES programme. The shaded area (in green) shows the acceptable range of the PD-SID metric. For each point, the error bars represent the standard deviation in the comparisons of all**

**pairs of sites. The number in the legend indicate the number of sites over France were the given profile is available.**

## 3.5 Long-term trends of PM$_{10}$ sources

Figure S10 in the SI presents the long-term trend of the observed PM$_{10}$ in the OPE site. The PM$_{10}$ levels appear to be consistent from 2012 to 2020 and there is no clear increasing or decreasing trend found in PM$_{10}$ ($r^2$=0.2, Table 1). However, there is a clear decline found in EC mass concentrations ($r^2$=0.9) with a reduction of 22 ng m$^{-3}$ year$^{-1}$ ($p \leq 0.01$) (Figure S11 in the SI).
This could indicate that the mass contribution of one or more sources contributing to EC should also be decreasing. Following Germany and Italy, France places third on highly impacted countries in Europe from vehicular exhaust emissions (Anenberg et al., 2010). Through the years, a variety of vehicular regulations have been adopted to reduce traffic-related emissions, not only in France (Bernard et al., 2020), but also across Europe (Wappelhorst and Muncrief, 2019). The data obtained in the OPE site poses an interesting opportunity as it covers 9 years of sampling in a rural area, making it possible to investigate emission
trends over a long period of time in a site representing a background atmosphere.

Using the model described in section 2.6 (Eq. 2), the STL deconvolution of the PMF-resolved sources in the OPE site were also investigated. It is extremely significant that the contributions from the traffic factor in the OPE site also decreased substantially, as presented in Figure 6. A very large reduction of 58% from year 2012 to 2020 based on average mass contribution and an overall yearly average reduction of 104 ng m$^{-3}$ y$^{-1}$ ($p \leq 0.01$) were found. In parallel, there is also a reduction
observed in the sulphate-rich factor, proposed as a highly anthropogenic-derived factor (see Figure S12 in the SI), with 66% reduction of average mass contribution and an over-all yearly average reduction of 72 ng m$^{-3}$ y$^{-1}$ ($p \leq 0.01$). Indeed, several other sources have shown lower but significant decrease of their mass contribution over the years (Table 1) in the OPE site, except for the fresh sea salt, nitrate-rich, and MSA-rich factors.

These findings allowed the unravelling of the decreasing trend in terms of source contributions by the STL model. The STL
deconvolution was applied on all the identified sources, which clearly showed that the traffic source has the highest tendency with a decreasing trend. The other major sources of PM, such as biomass burning, mineral dust, nitrate-rich sources, do not have as much decreasing tendency as the traffic factor. The internal annual variabilities of weather/climate conditions might not be the leading factors explaining these trends, as they would have affected PM sources in the same way.

**Table 1: STL tendencies of the observed PM$_{10}$ and each PMF-resolved source contributions to PM$_{10}$ from year 2012 to 2020 in the**
**OPE site.**

| | Tendency (µg m$^{-3}$ year$^{-1}$) | $r^2$ | $p$-value |
|---|---|---|---|
| Observed PM$_{10}$ | -0.107 | 0.21 | $\leq 0.01$ |
| EC | -0.022 | 0.89 | $\leq 0.01$ |
| Traffic | -0.104 | 0.67 | $\leq 0.01$ |
| Aged sea salt | -0.042 | 0.52 | $\leq 0.01$ |
| Fresh sea salt | 0.052 | 0.77 | $\leq 0.01$ |
| Mineral dust | -0.055 | 0.09 | $\leq 0.01$ |
| MSA-rich | 0.013 | 0.14 | $\leq 0.01$ |
| Nitrate-rich | -0.002 | 0.00 | 0.90 |
| Primary biogenic | -0.027 | 0.82 | $\leq 0.01$ |
| Sulphate-rich | -0.072 | 0.57 | $\leq 0.01$ |
| Biomass burning | -0.033 | 0.64 | $\leq 0.01$ |

The downward trends found in our study are well consistent with other existing studies in Europe (Li et al., 2018; Sun et al., 2020; Salvador et al., 2012; Pandolfi et al., 2016; Gama et al., 2018; Amato et al., 2014), nearly all of them conducted in urban areas. Pandolfi et al. (2016) found a significant long-term decrease of the contributions from anthropogenic emissions

(specifically a mixed industrial/traffic factor, -0.11 μg m$^{-3}$ year$^{-1}$, 56% total reduction) in a regional background site in altitude in northeast of Spain (Montseny, Spain) from 2004 to 2014. This is also consistent with a similar study in the metropolitan area of Madrid, Spain (Salvador et al., 2012) which showed a reduction of 32.7% attributed to traffic emissions, alongside the decrease of the carbonaceous and $SO_4^{2-}$ in PM. In a southern Spain area (Andalusia), the same group also found a consistent decreasing trend of PM at some traffic and urban sites in the region (Amato et al., 2014).

Another long-term study in Central Europe (Sun et al., 2020) focusing on equivalent black carbon (eBC) concentrations found decreasing trends in high-altitude Alpine sites located in Germany (-3.88% year$^{-1}$, [-10.15%, 0.56%]) and Switzerland (-3.36% year$^{-1}$, [-8.71%, -0.28%]). These findings are also consistent with results from other parts of Europe, with the largest decrease found in OC up to -48% (Cusack et al., 2012) and the decrease in PM has been associated to non-meteorological factors (Barmpadimos et al., 2012). Other studies with pluri-annual series of data on PM chemistry in rural environments in Europe

includes Spindler et al. (2013) (Melpitz, Germany, including EC measurement for 2003-2011), and Grange et al. (2021) (Payern, Switzerland comparison of 3 periods every ten years since 1998, including EC and trace elements). Both are showing decrease of EC concentrations over time during the study. Finally, while these studies did not target specific chemical species solely linked to vehicular emissions, most of them attributed the decline to the efforts to reduce vehicular emissions and other mitigation policies in their respective areas.

It should be noted that the role of meteorology on the observed decrease in PM in these studies (including ours) cannot be totally ruled out (Hou and Wu, 2016; Czernecki et al., 2017; Kim, 2019) and is generally not fully considered. In most cases, there is a complex interplay between PM and meteorological conditions that could increase or decrease PM mass concentration (Chen et al., 2020). Indeed, there are some studies at high-altitude or regional background sites that highlighted a concurrent role of changing meteorology and changes in frequency of Saharan dust advections to Europe (Brattich et al., 2020) in

modulating the dust concentrations in the atmosphere. The study at Melpitz (Spindler et al., 2013), despite an in-depth work on the wind sector classification, does not address the impact of possible changing in the air mass origin on long-term changing origins.

The evolution of the absolute concentration of the traffic factor in μg m$^{-3}$ at the OPE site was also compared to an evaluation of black carbon (BC) emissions in kilotons by the transport sector for overall France, provided by the CITEPA, the official

agency in charge of the emissions inventory in France (https://www.citepa.org/fr/2021-bc/). Both series were converted in an arbitrary level starting from 100, using 2012 as the base year (Figure 7). This figure shows an excellent agreement in the trend and in the total percentage decrease (%) for estimated BC emissions from traffic (-64%) and the traffic source contributions observed at the OPE site (-52%), between the years 2012 to 2020. While local or regional changes in meteorology may be a factor in the evolution of the concentrations observed, this is unlikely to be the dominant one in the evolution of the

concentrations of chemical species related to traffic emissions, in light of the strong correlation observed with the national emissions inventory in France.

The interesting points of our study is that it pertains directly to a specific source, identified with a long term and robust PMF study. Further, as there are minimal local anthropogenic sources expected in the OPE site, it may be safe to assume that these contributions to PM mass in this area are influenced by LRT of pollutants. Our results indicate that the implementation of

emission control policies are also playing a role in the consistent decrease in traffic emissions in rural sites far away from direct emissions.

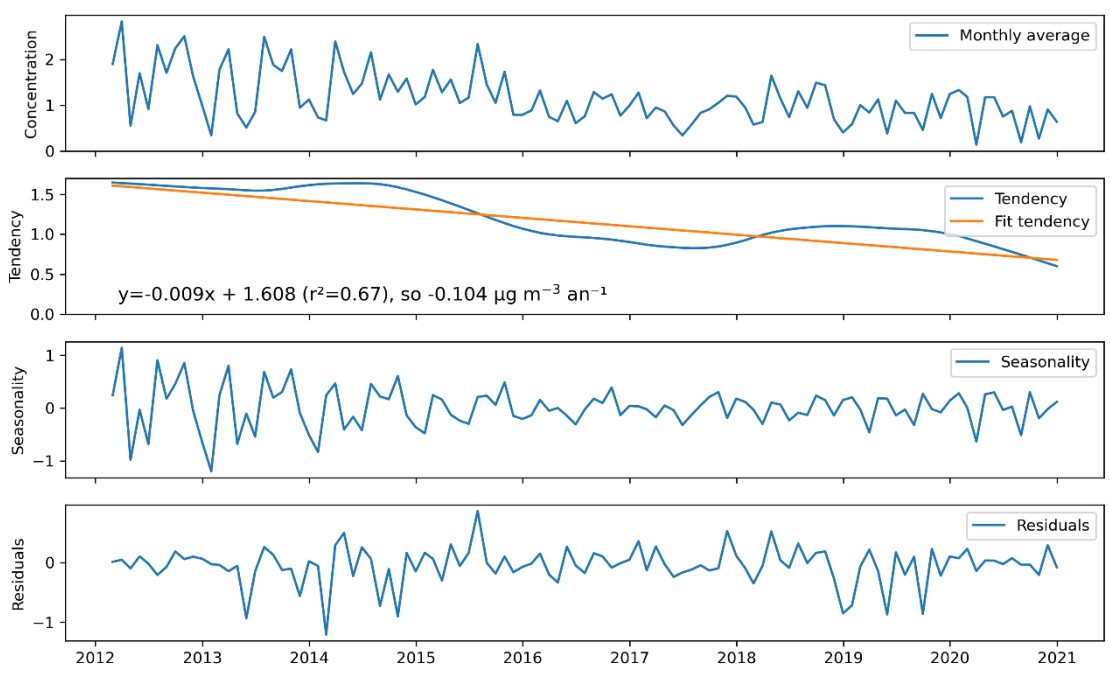

**Figure 6: The Season-trend (STL) deconvolution of contributions in µg m$^{-3}$ from the traffic factor to PM$_{10}$ from year 2012 to 2020.**

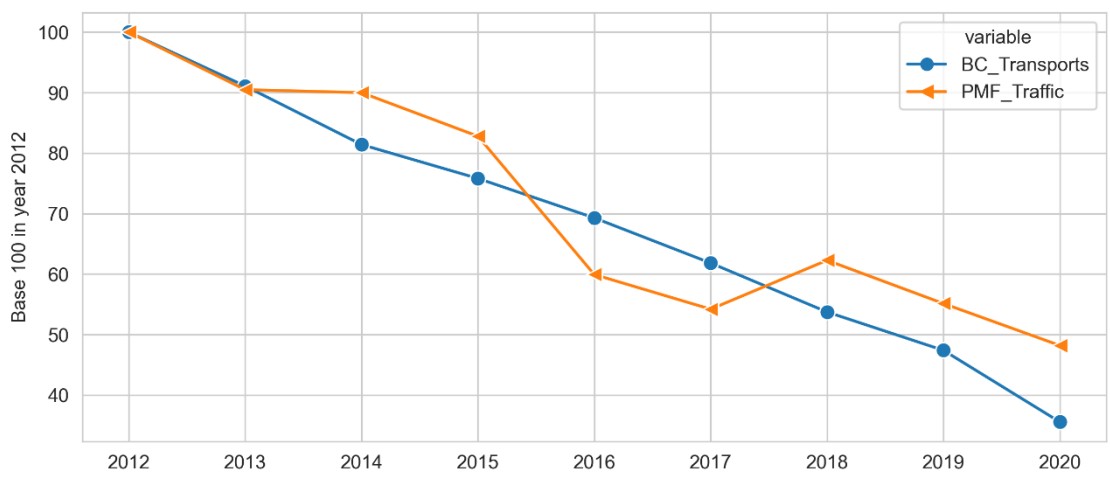


**Figure 7: Comparison of the evolution of the traffic factor source contribution (in µg m⁻³) at the OPE site and the black carbon (BC) emissions (in kilotons) by the transport sector (source: CITEPA, https://www.citepa.org/fr/2021-bc/) for overall France.**

### 3.6 Temporal trends of observed OP of PM₁₀

Figure 8 presents the observed average values of OP of PM$_{10}$ in the OPE site compared to the other sites in France (Calas et al., 2018, 2019; Weber et al., 2021, 2018). All series cover at least one year of sampling. As expected, the OP level in a rural background is much lower (about 2 to 8 times) than other typologies including traffic, urban, and urban alpine sites, for both AA and DTT assays. Further, the average ratio between urban sites and the rural OPE site is generally much higher for OP's than for PM mass, an indication that the nature of the particles at the rural site make them less oxidant than in urban areas, as

already pointed out in Daellenbach et al. (2020).

Figure 9 presents the daily and monthly mean distributions of observed PM$_{10}$ and OP activity ($OP_v^{DTT}$ and $OP_v^{AA}$) from June 13, 2017 to December 22, 2020. There is an observed seasonality, where PM$_{10}$ and $OP_v^{DTT}$ appear to be similar with relatively higher levels during warmer months. On the contrary, $OP_v^{AA}$ has slightly higher levels during colder months.

In many European studies, the seasonality in PM$_{10}$ mass concentration can be usually explained by higher contributions from

biomass burning during winter (Bessagnet et al., 2020; Tomaz et al., 2017), especially in alpine valleys (Calas et al., 2019b; Favez et al., 2010; Herich et al., 2014; Srivastava et al., 2018; Tomaz et al., 2016, 2017; Weber et al., 2018, 2019; Borlaza et al., 2021). Similarly, this seasonal pattern has been observed in OP as well (Borlaza et al., 2021; Weber et al., 2018; Calas et al., 2019b; Weber et al., 2021). However, the typology (i.e., rural) of OPE site could be associated with a different type of OP temporal profile as it is far from direct anthropogenic emission sources (but not from vegetation and soil biogenic emissions).

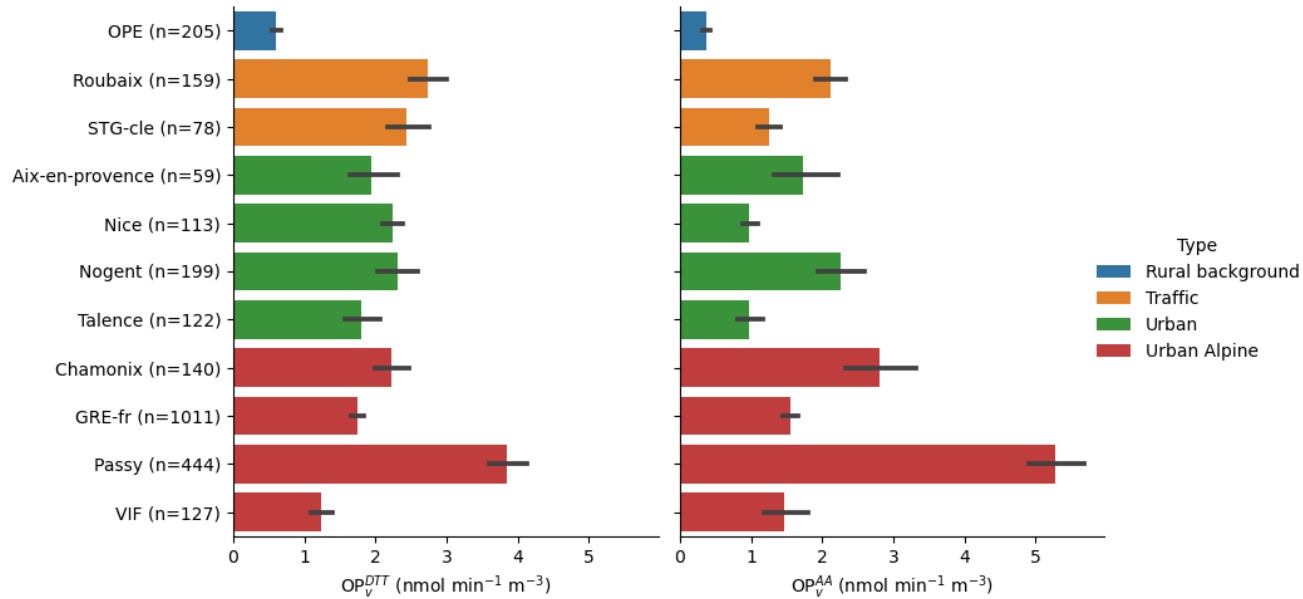

**Figure 8: The comparison of observed OP activity ($OP_v^{DTT}$ and $OP_v^{AA}$) between the OPE site and other sites in France. Bar plots depict the mean value with standard deviation.**

With the large influence of LRT on the sources of PM in the OPE site, this also poses an opportunity to investigate on the impact of LRT on OP. In fact, few studies have looked into the effects of aerosol aging to OP properties (Guascito et al., 2021; Bates et al., 2019). Pietrogrande et al. (2019) reported association of OP to redox-active organics linked to photo-oxidative aging. Using backward trajectory analysis, Wang et al. (2020) found strong effects of LRT on OP of fine PM. This is also consistent with the findings in Paraskevopoulou et al. (2019) which revealed highly oxygenated secondary aerosols as one of the main drivers of OP in fine PM, further highlighting the importance of combustion and aging processes in OP. In a shipborne measurement study in South Korea, a higher intrinsic OP has also been found in samples where secondary aerosol formation is more dominant, also suggesting strong impacts of long-range transported PM (Oh et al., 2021). Cesari et al. (2019) found negligible contribution from secondary sulphates, but have found relevant OP contributions from a factor identified as a combination of vehicular traffic and secondary nitrates. All these studies used DTT assay to measure the OP of PM.

The OP assay sensitivity to specific species and/or sources can also explain the difference in seasonality found in $OP_v^{DTT}$ and $OP_v^{AA}$ in our study. $OP_v^{DTT}$ appears sensitive towards organics, metals, and, possibly, a synergistic effect between the two (Bates et al., 2019; Dou et al., 2015; Fang et al., 2017; Gao et al., 2020b, a; Jiang et al., 2019; Weber et al., 2021; Yu et al., 2018; Borlaza et al., 2021), while, $OP_v^{AA}$ shows sensitivity mostly towards metal species (Bates et al., 2019; Crobeddu et al., 2017; Visentin et al., 2016; Weber et al., 2021; Borlaza et al., 2021).

Generally, current literature poses importance of the role of secondary/aged aerosols on OP of PM, especially those with influence from anthropogenic sources. However, to our knowledge, this is the first long-term OP study, especially dealing with ambient samples far from direct emissions at a rural background site.

### 3.7 Sources of OP in PM$_{10}$

The sources of OP in PM$_{10}$ was apportioned following an OP deconvolution method proposed by Weber et al. (2018) using the source contributions (µg m$^{-3}$) obtained in the PMF and the measured OP (nmol min$^{-1}$ m$^{-3}$) at the OPE site. Generally, the modelled OP ($OP_m$) is within range of the observed OP, with a reasonable reconstruction ($OP_v^{DTT}$ ($r$=0.76) and $OP_v^{AA}$ ($r$=0.76)).

The $OP_m$ of each PM source is given in Table 2, where $OP_m^{DTT}$ can range from -0.01±0.02 to 0.10±0.03 nmol min$^{-1}$ µg$^{-1}$ and $OP_m^{AA}$ can range from -0.001±0.02 to 0.16±0.03 nmol min$^{-1}$ µg$^{-1}$. Generally, higher $OP_m$ indicates higher redox-activity associated with the factor. There are some differences in the $OP_m$ based on the type of assay used and this can be attributed to the sensitivity of the assay towards certain redox-active species in PM (Borlaza et al., 2021; Xiong et al., 2017; Charrier and Anastasio, 2012).

In terms of overall daily mean contribution, as presented in Figure 10, the main contributors to PM$_{10}$ mass are the nitrate-rich, mineral dust, and sulphate-rich factors in the OPE site. However, in terms of $OP_v^{DTT}$, the mineral dust factor showed the highest average contribution (0.15 nmol min$^{-1}$ m$^{-3}$), followed by the sulphate-rich (0.11 nmol min$^{-1}$ m$^{-3}$) and traffic (0.07 nmol min$^{-1}$ m$^{-3}$) factors. For $OP_v^{AA}$, the biomass burning factor showed the highest contribution (0.12 nmol min$^{-1}$ m$^{-3}$), followed by the traffic (0.07 nmol min$^{-1}$ m$^{-3}$) and nitrate-rich (0.06 nmol min$^{-1}$ m$^{-3}$) factors.

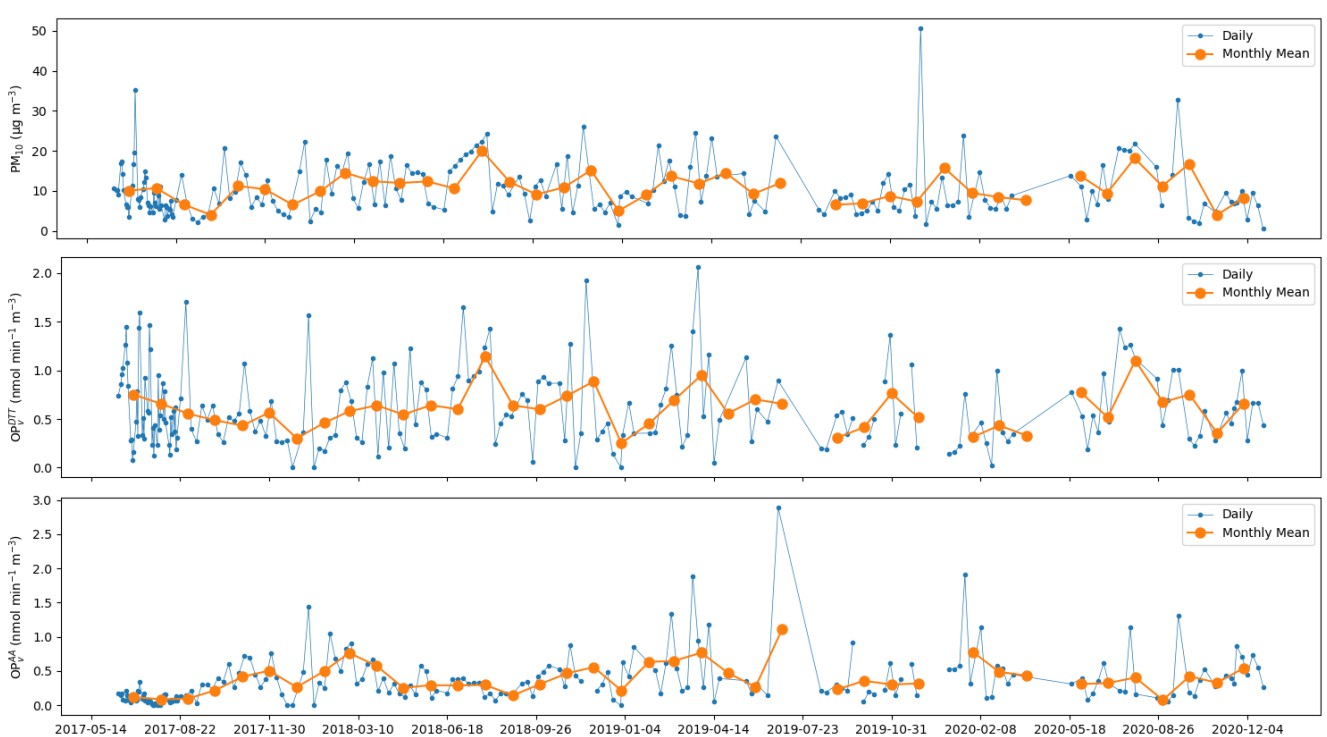


**Figure 9: Temporal distributions of observed PM$_{10}$ and OP activity ($OP_v^{DTT}$ and $OP_v^{AA}$) from year 2017 to 2020 in the OPE site in terms of daily and monthly mean.**

Although lower in magnitude, the OP contribution of mineral dust, traffic, and biomass burning (only in $OP_v^{AA}$) are also prominent in the OPE site, similar to other sites in France (Weber et al., 2021). These sources are commonly composed of

species that are highly redox-active, hence it is not surprising that they are one of the main drivers of OP even in a rural site. Both mineral dust and biomass burning are also sources that can be associated with LRT and aging, respectively, which are atmospheric processes linked to increased OP (Pietrogrande et al., 2019; Wang et al., 2020; Paraskevopoulou et al., 2019; Oh et al., 2021). Further, while the mineral dust profile in OPE is considered homogeneous with those determined in other parts of France, as discussed above, its chemical composition includes slightly larger fractions of some metals, particularly Fe and

Cu, possibly making it more redox-active.

**Table 2: Regression coefficients (i.e., intrinsic OP or $OP_m$) expressed in nmol min$^{-1}$ µg$^{-1}$ at the OPE site for the DTT and AA assays. The values are the mean ± standard deviation and the $p$-value is in the parenthesis.**

| Factor | $OP_m^{DTT}$ | $OP_m^{AA}$ |
|---|---|---|
| Traffic | 0.07±0.04 ($p$=0.09) | 0.08±0.01 ($p{\leq}0.01$) |
| Aged sea salt | 0.06±0.04 ($p$=0.20) | 0.08±0.04 ($p{\leq}0.05$) |
| Fresh sea salt | 0.02±0.02 ($p$=0.40) | 0.02±0.02 ($p$=0.22) |
| Mineral dust | 0.10±0.03 ($p{\leq}0.01$) | 0.04±0.01 ($p{\leq}0.01$) |
| MSA-rich | 0.04±0.04 ($p$=0.21) | -0.06±0.03 ($p$=0.06) |
| Nitrate-rich | 0.01±0.01 ($p$=0.10) | 0.03±0.01 ($p{\leq}0.01$) |
| Primary biogenic | -0.01±0.02 ($p$=0.68) | -0.005±0.01 ($p$=0.82) |
| Sulphate-rich | 0.10±0.03 ($p{\leq}0.01$) | -0.001±0.02 ($p$=0.98) |
| Biomass burning | 0.03±0.03 ($p$=0.42) | 0.16±0.03 ($p{\leq}0.01$) |
| *Intercept (nmol min$^{-1}$ m$^{-3}$)* | *0.14±0.06 ($p{\leq}0.05$)* | *0.07±0.05 ($p$=0.16)* |

As observed in other studies (Daellenbach et al., 2020; Borlaza et al., 2021; Weber et al., 2021), there is also a clear difference in source ranking when considering the PM mass or OP, highlighting that the sources driving PM mass are not the same as the ones driving OP activity. The mass contributions of the nitrate-rich factor can be twice that of the traffic factor. However, in terms of OP (both $OP_v^{DTT}$ and $OP_v^{AA}$), the traffic factor contribution can be twice as much as that of the nitrate-rich factor. The biomass burning factor, with only <1 µg m$^{-3}$ mass contribution on annual average, appears to have the highest contribution in

$OP_v^{AA}$.

Some previous studies associated secondary inorganic sources (SIA) with minimal contributions on PM toxicity (Cassee et al., 2013; Daellenbach et al., 2020; Park et al., 2018). However, the nitrate-rich factor apportioned in the OPE site showed contributions to both $OP_v^{DTT}$ and $OP_v^{AA}$, and the sulphate-rich factor to $OP_v^{DTT}$. In the sulphate-rich factor, a fraction of OC (5%) and metals (Se (42%), Zn (44%), Cu (27%), and Sb (25%)) were apportioned, while in the nitrate-rich factor there are

contributions from OC (6%), EC (4%), and metals (Sb (6%) and Sn (11%)). These species are commonly anthropogenic-derived, signalling that the sulphate- and nitrate-rich factors could be influenced by these types of emissions as well. In fact, a similar study considered that both SIA factors can be associated to anthropogenic SOA sources (Borlaza et al., 2020).

Although both DTT and AA assays represent potential PM-induced oxidative stress, through in vivo interactions between redox active components in $PM_{10}$ and biological oxidants, it can be observed that they differ in terms of source impacts. This can be attributed to the sensitivity of each assay to specific species and/or emission sources of PM. Nevertheless, most of the sources of PM suggested to be anthropogenic-derived or impacted, such as traffic, mineral dust, nitrate-rich, sulphate-rich (only in $OP_v^{DTT}$), and biomass burning (only in $OP_v^{AA}$), were all usually in upper half of the scale (Figure 8) in terms of $OP_v$ contributions. The knowledge of source-specific $OP_v$ contributions provide useful information on the main drivers of $OP_v$, even in a rural area such as the OPE site.

There is an interesting seasonality observed in the OP of PM, as previously shown in Figure 9. With the OP source deconvolution method, this seasonality has been revealed in terms of PM sources, as presented in Figure 11. During colder months, the biomass burning factor clearly dominated the $OP_v^{AA}$ contributions. During warmer months, the $OP_v^{DTT}$ contributions were dominated by the mineral dust factor. However, there is a relatively consistent monthly contribution for both assays coming from the traffic factor.

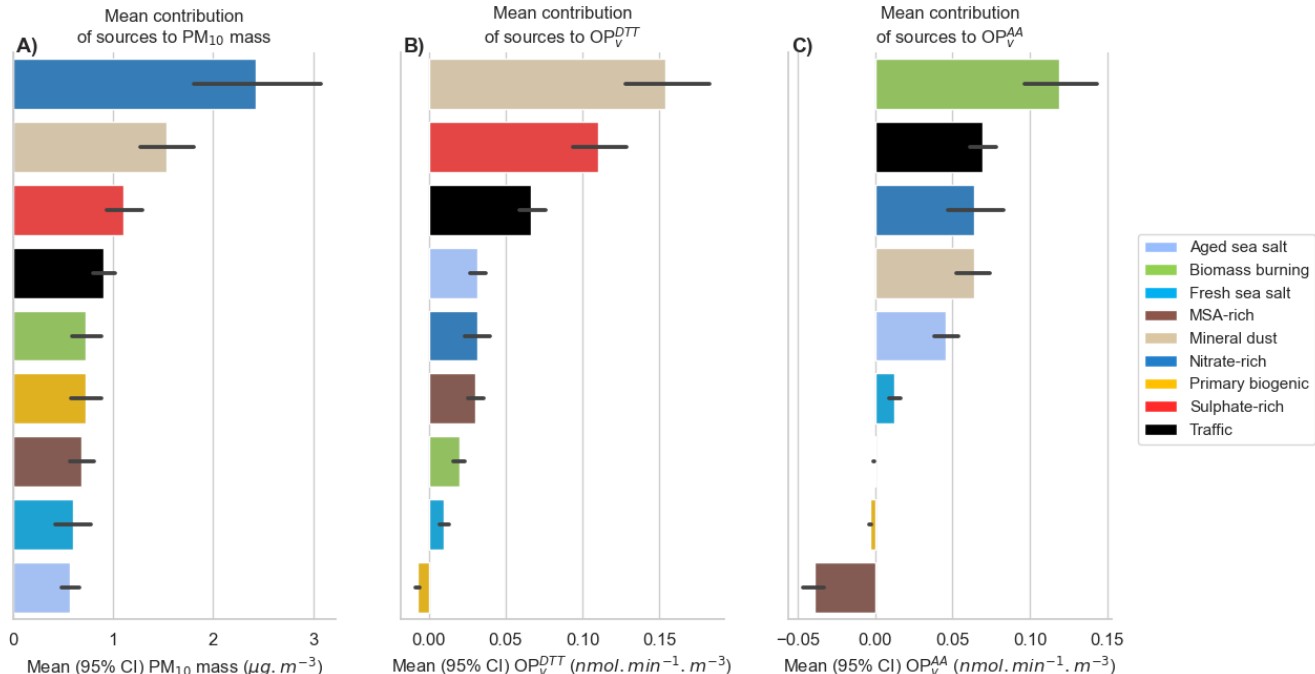

**Figure 10: Overall daily mean $OP_v$ contribution of the sources to $PM_{10}$ and OP activity ($OP_v^{DTT}$ and $OP_v^{AA}$) using MLR analysis in the form of mean and 95% confidence interval of the mean (error bar) ($n$=168 samples).**

It is also interesting to note the negative contributions from some sources. This negative contribution is brought by a negative intrinsic OP ($OP_m$ obtained in the OP deconvolution method (Table 2)). This can be broadly interpreted as follows: for every 1 µg m$^{-3}$ increase in the MSA-rich factor, there is an associated decrease in $OP_v^{AA}$ contributions ($OP_m$=-0.06±0.03, $p$=0.06). A

similar interpretation can be done for the primary biogenic factor and its $OP_v^{DTT}$ contributions ($OP_m$=-0.01±0.02, $p$=0.68). However, it is important to note that both MSA-rich and primary biogenic factors do not always present a negative $OP_m$ in every site investigated in our group. One cannot completely assume that these two factors always act as suppressors of OP of PM. In fact, these two factors have shown high $OP_m$ variabilities across different sites in France (Weber et al., 2021). With the use of fit-for-purpose organic tracers, possible mixing issues in these factors can be minimized (Borlaza et al., 2021). However, these supplementary tracers were not available in the OPE site, making it difficult to eliminate potential influence from other factors or species in PM.

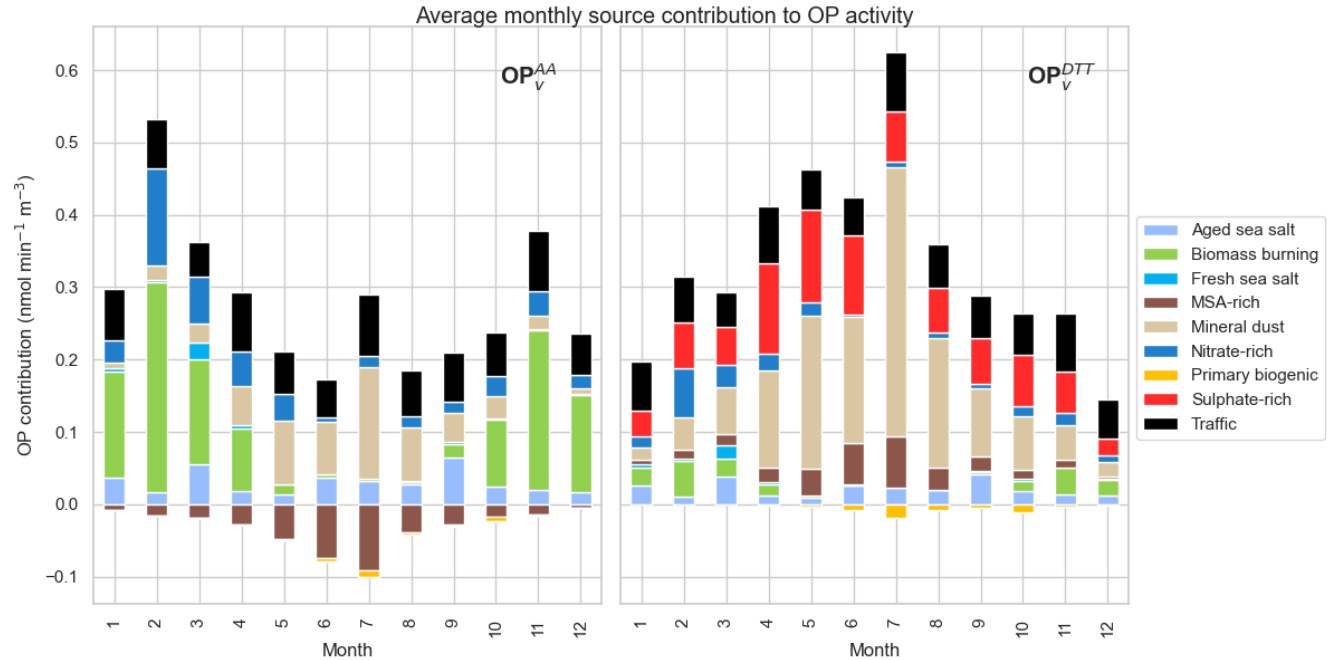

Figure 11: Monthly contribution of sources to OP activity in nmol min⁻¹ m⁻³ ($OP_v^{DTT}$ and $OP_v^{AA}$) from year 2017 to 2020 in the OPE site. Note: The months are labelled from January (1) to December (12).

Figure 12 presents the yearly contribution of each factor to the OP of $PM_{10}$ for the 3 years investigated in this study. There is no clear decreasing trend in the total OP reconstructed by MLR analysis. Although there is a decreasing trend found in the mass contributions of the traffic factor, this was not clearly reflected in its OP contributions. The other sources seem to have comparatively consistent OP contributions from 2017 to 2020, and no notable tendency can be found in the total OP contribution as opposed to the contributions to $PM_{10}$ mass showing decreasing trends. This may be explained by the limited subset of samples for the OP assay (OP data spanning 4 years against 9 years for the PMF data), the shorter time range being insufficient to reach significance and robustness in the trend assessment of OP levels.

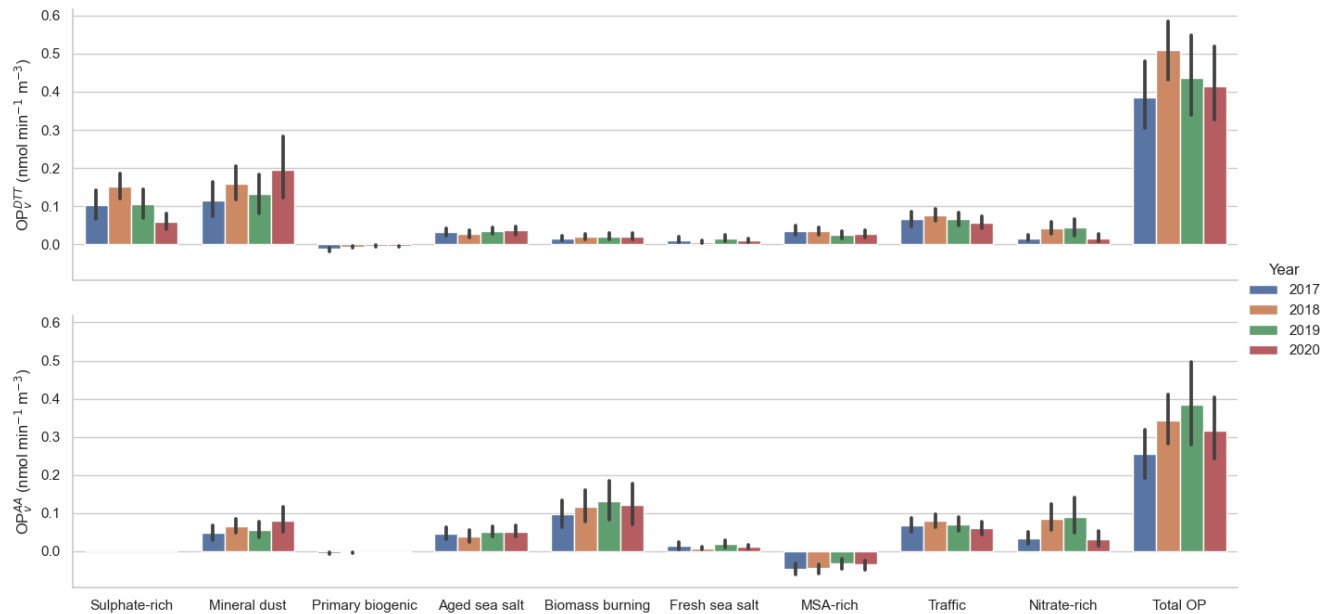

**Figure 12: Yearly average contributions of sources to OP activity in nmol min⁻¹ m⁻³ ($OP_v^{DTT}$ and $OP_v^{AA}$) from year 2017 to 2020 in the OPE site. Note: Total OP is the sum of OP contributions of all sources as modelled by the MLR analysis.**

### 3.8 Limitations of the study

In spite of the advantages offered by the long-term monitoring in the OPE site, there were a few limitations encountered during the investigation of the dataset. Each limitation is discussed as follows:

- There was a change in sampling duration between the collection performed in year 2012 to 2016 (7-day sampling) and 2016 to 2020 (24-hour sampling). A 7-day filter sample includes both weekdays and weekends, whereas a 24-hour sample will either be a weekday or weekend, depending on the sampling interval. This implies that the weekly collected samples may contain particles that are not fully captured in a daily sample. However, since the OPE site is quite distant from direct emissions, the expected difference in the weekday and weekend levels should be relatively
small. Further, PMF source apportionment were conducted separately (i.e., 7-day samples versus 24-hour samples), leading to very similar results for the chemical profiles and source contributions, justifying the coupled analysis. (see SI, Table S4 and Figures S13 to S21)

- The lack of fit-for-purpose tracers to fully elucidate the influence of SOA in a site with relevant distance from direct emissions (or rural typology) such as OPE. In Borlaza et al. (2021), a secondary biogenic oxidation source was
additionally identified using organic tracers (3-MBTCA and pinic acid), while anthropogenic influence was supported by contributions of phthalic acid, even in secondary aerosol sources. With the typology in the OPE site, this would have been useful.

- The use of a single chemical profile for long-term PM source apportionment could be limiting for the PMF model. As we have found consistent decrease in some species, particularly EC, perhaps a rolling PMF (e.g., yearly PMF) could better capture possible changes in the source profiles.

- The absence of samples for OP analysis from years prior to 2017 have limited the investigation of long-term OP in the OPE site. Consequently, it was not easy to capture the decrease in OP contributions from the traffic factor as similarly captured in the mass contributions. Perhaps a hindcasting method on the years without OP data could have been performed, however that would heavily rely on the $OP_m$ modelled from years 2017 to 2020, which can lead to bias in the results.

- In the year 2020, a series of lockdown restrictions were placed nationally as a response to the coronavirus disease (COVID-19) pandemic. In the OPE site, there is no clear decrease in average $PM_{10}$ mass concentration in 2020 (Figure 12) that could have greatly affected the results of this study. In fact, excluding all samples from year 2020, the traffic factor contributions to $PM_{10}$ still has a reduction of 39% from year 2012 to 2019 and an overall yearly average reduction of 135 ng m$^{-3}$ y$^{-1}$ ($p \leq 0.01$).

**4 Conclusions**

Over the 9-year analysis in a rural background site in France (OPE), the observed $PM_{10}$ mass concentration and OP were found to be much lower than other sites in France. The sources of $PM_{10}$ mass and OP were apportioned using PMF and MLR analysis, respectively. The nine identified factors relevant for $PM_{10}$ include secondary inorganics (nitrate- and sulphate-rich), traffic, mineral dust, biomass burning, sea salts (fresh and aged), primary biogenic, and MSA-rich.

A redistribution of the factor impacts between mass and OP contributions was observed, underlining the importance of taking into account the redox activity of PM when considering their potential health effects. Based on PM mass, the major contributors are nitrate- and sulphate-rich factors, both factors being associated with secondary inorganics formed during long range transport (LRT). On the other hand, based on OP activity, the main contributors are mostly anthropogenic-derived sources such as traffic, mineral dust, and biomass burning factors.

As the OPE site is located far from direct anthropogenic emissions, the influence of LRT processes was noted in some sources. Sources such as sulphate- and nitrate-rich, MSA-rich, and aged sea salt factors have shown potential mixing with other anthropogenic sources, most probably due to the transit time. These potential mixing and aging processes were reflected in the chemical mass profile of each factor as well as in their OP contributions.

Thanks to the long-term dataset in the OPE site, it was observed that the traffic factor contribution to total $PM_{10}$ has decreased over the years for this site that may well represent the French national background PM. This decrease is much larger than any change observed for the other PM sources and is in excellent agreement with estimations in the decrease in BC emissions from the transport sector all over France from the national inventory. This effect may be attributed to improvement of the exhaust emission of terrestrial transportation fleet, and/or to regulations restricting vehicular emissions in bigger cities and/or other

regional-scale. However, persistent changes in meteorological conditions influencing the transport of air masses to OPE or formation of PM during this transport cannot be totally ruled out.

**Acknowledgements**

The authors wish to thank all the many people from the different laboratories (OPE, IGE, Air O Sol analytical platform, LCME) and from the regional air quality monitoring network Atmos Grand Est, who actively contributed over the years in filter
sampling and/or analysis. The data used for the comparison in Figure 8 is obtained from many different programs, including the CARA program coordinated by O Favez (Favez et al., 2021).

**Code availability**

The software code could be made available upon request by contacting the corresponding author.

**Data availability**

The chemical, PMF, and OP datasets could be made available upon request by contacting the corresponding author.

**Author contributions**

SC manages the overall observatory at OPE, including the supervision of the PM sampling. JLJ designed the project, in collaboration with SC. LB, SW, and AM did the curation of the data base. LB performed the data analysis and wrote the paper. GU manage the OP analytical procedures at IGE. VJ designed part of the analytical method on the Air O Sol plateau. MC is
the representation of Atmo GE who helped maintaining the sampling equipment. All authors read and commented the manuscript.

**Competing interests**

The authors declare that they have no conflict of interest.

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
