# Peer review of "9-year trends of $PM_{10}$ sources and oxidative potential in a rural background site in France"

_Atmospheric Chemistry and Physics, 2021_

## Referee Comment (RC1)

**Revision of "9-year trends of PM10 sources and oxidative potential in a rural background site in France" by Borlaza L.J. et al.**

**General comments**

This work presents the results of long-term monitoring and characterization of sources of PM10 and their contribution to the Oxidative Potential at a rural French site. The work is interesting and generally well written. However, the paper could benefit of some revisions.

First of all, the analysis of the variability of PM10 concentration (Section 3.1) focuses on just reconstructed PM10, and it is not clear how much of the measured mass is efficiently reconstructed. In addition, the analysis covers only yearly averages and therefore interannual variability, while a focus on seasonal and perhaps subseasonal time scales could be interesting as well.

Secondly, the information reported on the choice of the PMF solution is not complete, and as such it is not possible to judge if the choice was done appropriately.

As a third point, the STL analysis is applied to all factors, but Figure 6 focuses only on the traffic factor: however, it is interesting to note that papers analysing long term trends at high-altitude or regional background sites (in some cases even less impacted by anthropogenic sources than this particular site) have indicated an important role of changes in meteorology for the observed decrease in PM10 in the last decades. This would be very interesting to analyse here as well, because it could indicate that the role of control policies in driving PM10 decreases was sustained by meteorological changes. As such, this investigation could complement nicely the findings presented here.

Below I present an additional list of specific suggestions to improve the overall quality of the work.

**Specific comments**

Pay attention to similarity matrix and check if you can rephrase parts that look similar to other articles.

Line 15: A specification of the name/location of the site could be given in addition.

Line 17: Change "from" to "analysed on".

Lines 19-22: The difference between the two sentences is not straightforward and clear. Could you please rephrase and make the difference clearer?

Line 24: But this is not a result of this study, since you did not analyse urban areas.

Line 25: Change "However, this" to "even though this"

Line 26: Change "signal" to "indicate".

Line 29: This sentence is not clear: revise.

Line 30: Change "on chemical characterization and sources of PM" to "PM chemical characterization and sources" and "is concerned" to "focuses".

Line 31: Delete "as they are the places".

Line 33: Change "done" to "carried out" and delete "to try".

Line 34: Change "could" to "can".

Line 35: "geochemical" may be not the most appropriate term in this context. In addition, the pollutants are transported, not the sources. Please revise.

Lines 34-35: This sentence is not clear: could you please rephrase it?

Lines 36-37: Well, not only large-scale processes, but also mesoscale processes are needed for chemical transport models. Revise.

Line 38: Again, "geochemical" is not the appropriate term in this context.

Lines 43-44: Please rephrase as "However, only few sites provide long-term in-depth series of PM chemical speciation data."

Line 46: Delete "would".

Lines 47-48: Change "the case of the oxidative potential (OP) of PM" to "the case of PM oxidative potential (OP)"

Line 52: Change "see" to "analyse" or "investigate"

Line 53: Add "the" before "efficiency".

Line 55: Change "measurement" to "measurements".

Line 56: Change "large filter" to "long-term filter"

Lines 55-60: I would suggest presenting the structure of the work rather than summarizing the results and conclusions.

Line 63: The acronym was introduced previously without explanation: better move this explanation to the first time it is cited.

Line 66: Change "a good" to "considered representative".

Lines 63-68: Any additional description of the typical local climate? This can affect PM concentrations and may be relevant for the rest of the discussion.

Line 67: Add "at this site" after "chemistry".

Lines 75-78: This means that you analysed only field blanks and not blank filters? With which frequency did you analyse these field blanks?

Line 89: What is this "range of ratio"? Please explain better.

Lines 83-111: Any specifications of the Limits of Detection, and other experimental parameters?

Line 109: Change "includes" to "including".

Lines 115-116: If the analysis was started on samples collected from June 13, 2017 to December 22, 2020, it means that then you analysed also the rest of the samples. Is this true? If not, please revise.

Line 139: Change "PMF 5.0" to "EPA PMF 5.0".

Line 140: This definition is not correct: please revise.

Lines 147-148: Which paper did you follow for this step? And do you know that this is not complete to characterize the strength of the variables? The analysis of the residuals should be also made. In addition, how did you treat the additional uncertainty? Please revise.

Lines 150-152: How did you analyse weighted residuals? Please provide additional details.

Line 159: Change "in" to "with".

Line 164: Change "difference" to "differences".

Lines 164-171: Did you use any particular software for this calculation?

Line 182: Add "used" after "that". It is not clear if there are any differences with that methodology or not.

Line 201: Change "discusses" to "discuss" (twice).

Line 204: What does it mean "reconstructed"? How far is this reconstruction from the measured value?

Line 205: Is the value after the "±" the standard deviaton? Please specify.

Line 210: Delete comma after "although".

Lines 211-212: A decrease during this period is quite evident, while the change in composition is less evident.

Line 215: Is this a mean value? From Figure 2 this value does not seem constant.

Lines 219-220: This example does not explain much. Please revise.

Lines 220-221: This sentence is not well linked with the previous results shown. Please revise.

Lines 223-225: Please provide additional details on how this solution was selected. The signs of instability are worrying.

Lines 233-235: Couldn't you use a value more appropriate for a rural site?

Line 249: Change "lead" to "have led".

Lines 263-265: But local sources cannot be excluded: if not, we would have always higher nitrate and sulphate concentrations at rural sites than at urban ones, which is not the case.

Line 270-271: Can you explain the reasons of this seasonality in these 2 factors?

Line 281: Change "were" to "was":

Lines 296-297: Couldn't this be due to the fact that Na+ and Mg2+ are primary seasalt particles while MSA particles are secondary? What about the correlation with nss-SO42-? See for instance papers from the group of Silvia Becagli and Roberto Udisti (e.g., Udisti et al., 2016; Becagli et al., 2019, 2021). Did you try to analyse the source (with wind or back-trajectories)?

Line 305: Change "typologies" to "sites".

Line 332: Change "dissimilarity" to "dissimilarities".

Figure 6: Check the y-scale (measurement unit).

Lines 371-382: There are also studies at high-altitude or regional background sites, some of which highlighted a concurrent role of a changing meteorology and of a change in the frequency of Saharan dust advections to Europe. Please look at: Tsyro et al., 2018; Colette et al., 2011, 2017; Brattich et al., 2012, 2020 and references therein). Thus this discussion may be improved.

Figure 8: Please check the x-axis and add more ticks. There are some peaks in all series: did you analyse the presence of outliers and investigate their causes?

Line 457 and 458: Delete "as much as".

Line 475: Check the reference to the Figure.

Line 476: Add "in" before "Figure".

Line 482: Change "confident" to "confidence".

Line 484: Change "follow" to "follows".

Lines 511-514: This is not clear: the problem with weekly samples should be that differences and transient events (e.g., Saharan dust, fires, …) are smoothed but I cannot understand what you mean by "the weekly collected samples may contain particles that are not fully captured in a daily sample".

Line 530: Figure 11 is not about PM concentrations, but on OP contributions. A decrease in PM concentration could instead be observed as previously noted.

Line 546: Change "this" to "the".

Lines 548-550: As previously noted, this discussion is limited since there are studies evidencing a simultaneous effect of the changing meteorology. This point should be improved.

Code and data availability: Please check the statement for this: as on the ACP website https://www.atmospheric-chemistry-and-physics.net/policies/data_policy.html, "Authors are required to provide a statement on how their underlying research data can be accessed. … If the data are not publicly accessible, a detailed explanation of why this is the case is required. … Data do not comprise the only information which is important in the context of reproducibility. Therefore, Copernicus Publications encourages authors to also deposit software, algorithms, model code, video supplements, video abstracts, International Geo Sample Numbers, and other underlying material on suitable FAIR-aligned repositories/archives whenever possible"

---

## Author Comment (AC1)

**9-year trends of PM$_{10}$ sources and oxidative potential in a rural background site in France**

Authors' response

We would like to thank the referees for their time to evaluate our manuscript and for their positive and constructive feedbacks, which helped improve the quality of the paper. Our response to the comments are presented below (in blue):

**General revisions**: All grammatical and cross-referencing errors in the text were corrected (listed below). Thank you very much to our referees.

- Line 17: Change "from" to "analysed on".
- Line 25: Change "However, this" to "even though this"
- Line 26: Change "signal" to "indicate"
- Line 30: Change "on chemical characterization and sources of PM" to "PM chemical characterization and sources" and "is concerned" to "focuses"
- Line 31: Delete "as they are the places"
- Line 33: Change "done" to "carried out" and delete "to try"
- Line 34: Change "could" to "can"
- Lines 43-44: Please rephrase as "However, only few sites provide long-term in-depth series of PM chemical speciation data."
- Line 46: Delete "would"
- Lines 47-48: Change "the case of the oxidative potential (OP) of PM" to "the case of PM oxidative potential (OP)"
- Line 52: Change "see" to "analyse" or "investigate"
- Line 53: Add "the" before "efficiency"
- Line 55: Change "measurement" to "measurements"
- Line 56: Change "large filter" to "long-term filter"
- Line 66: Change "a good" to "considered representative"
- Line 67: Add "at this site" after "chemistry"
- Line 93: Define MSA
- Line 109: Change "includes" to "including"
- Line 139: Change "PMF 5.0" to "EPA PMF 5.0"
- Line 159: Change "in" to "with"
- Line 164: Change "difference" to "differences"
- Line 182: Add "used" after "that". It is not clear if there are any differences with that methodology or not. Response: The sentence was revised as follows: "This methodology is based on the procedure proposed in Weber et al. (2018)."
- Line 201: Change "discusses" to "discuss" (twice)
- Line 210: Delete comma after "although"
- Line 226: Remove 100 out of 100
- Line 249: Change "lead" to "have led". Response: The sentence was revised as follows: "As OC in a rural site can undergo multiple re-transformations in the atmosphere from the emissions sources, this has led to a wide range of OC-to-EC ratios as similarly found in Weber et al. (2019), hence this constraint was excluded."
- Line 281: Change "were" to "was"
- Line 305: Change "typologies" to "sites"
- Line 332: Change "dissimilarity" to "dissimilarities"

- Line 457 and 458: Delete "as much as"
- Line 476: Add "in" before "Figure"
- Line 482: Change "confident" to "confidence"
- Line 484: Change "follow" to "follows"
- Line 546: Change "this" to "the"

**Response to anonymous referee #1:**

Referee comment: First of all, the analysis of the variability of PM10 concentration (Section 3.1) focuses on just reconstructed PM10, and it is not clear how much of the measured mass is efficiently reconstructed. In addition, the analysis covers only yearly averages and therefore interannual variability, while a focus on seasonal and perhaps subseasonal time scales could be interesting as well.

Response: Thank you for the feedback. We have improved Figure 2 by adding the portion of unknown species in $PM_{10}$. It should be noted that only 69% ($n$=299) of the collected filters were paired with TEOM-FDMS measurements. Overall, the average reconstructed mass of non-volatile species measured on the filter, at about a grand average of 10% over the sampling period, is in the range of results from other studies in rural areas (e.g., Pey et al., 2009).

The authors acknowledge that our very large dataset could be worked in many different directions, including investigations of daily and seasonal evolution, and that all of these would give interesting information. However, this would also lead to a much longer paper, and we chose here to concentrate on 3 directions (sources of PM studied with PMF, sources of oxidative potential, and trends in the sources contributions), which covers already a large scope of investigation.

However, some information in the SI, like Figure S10, presents the STL deconvolution of $PM_{10}$ concentrations in terms of monthly and seasonal averages. The STL deconvolution (Seasonal and Trend decomposition using Loess) presented in this manuscript is a versatile and robust method for decomposing time series developed by Cleveland et al. (1990).

A paragraph in section 2.2 was also added and now reads as:

The PM10 measurements from the tapered element oscillating microbalance (TEOM-FDMS) are all in a daily (24-hour, 09:00 to 09:00) resolution, while the reconstructed PM10 were obtained from chemical analysis performed on filters collected on a weekly (7 days, 09:00 to 09:00) or daily (24-hour, 09:00 to 09:00) basis. A total of 299 out of 434 (69%) TEOM-FDMS measurements were paired with reconstructed PM10 data, due to many interruptions in the TEOM-FDMS functioning, in order to evaluate the semi volatile mass missing in the mass reconstruction with filter chemistry.

A paragraph in section 3.1 was also added and now reads as:

The yearly average volatile mass (i.e., unaccounted by chemical analysis), deduced from the difference between TEOM-FDMS measurements and reconstructed $PM_{10}$, ranges from 9% to 44% with an average of 22% (of the yearly median) and is well within range generally found in a rural environment (Pey et al., 2009).

Referee comment: Secondly, the information reported on the choice of the PMF solution is not
complete, and as such it is not possible to judge if the choice was done appropriately.
Response: We revised section 2.4.2 (Criteria for a valid solution) to elaborate more on the
specific conditions evaluated. This paragraph now reads as:
Solutions with a total number of factors from 6 to 11 were tested for the baseline models.
Following the recommendations of the European guide on air pollution source apportionment
with receptor models (Belis, 2019), the $Q/Q_{exp}$ ratio (<1.5), the geochemical interpretation of
the factors, the weighted residual distribution, and the total reconstructed mass were evaluated
during factor selection.
Moreover, the bootstrapping method (BS) was used on the final solution to estimate errors and
ensure the stability and accuracy of the solutions. The BS method was applied with 100
iterations of the model and contribution uncertainties are presented in the SI (S3) as mean±std
of the 100 BS runs. The contribution uncertainties were estimated based on the method
presented in Weber et al. (2019) and presented in Figures S2 to S10. The daily specie
contributions are estimated using:

$$X_{BSi} = G_{ref} \times F_{BSi}$$

where $F_{BSi}$ is the profile of the bootstrap i, and $X_{BSi}$ is the time series of each species according
the reference contribution $G_{ref}$ and the bootstrap run $F_{BSi}$.
Finally, the factor chemical profiles obtained during this study were compared with those from
previous studies in France, using the PD-SID method (Belis, 2019; Weber et al., 2019), in order
to validate their proper similarity.
Referee comment: As a third point, the STL analysis is applied to all factors, but Figure 6
focuses only on the traffic factor: however, it is interesting to note that papers analysing long
term trends at high-altitude or regional background sites (in some cases even less impacted by
anthropogenic sources than this particular site) have indicated an important role of changes in
meteorology for the observed decrease in PM10 in the last decades. This would be very
interesting to analyse here as well, because it could indicate that the role of control policies in
driving PM10 decreases was sustained by meteorological changes. As such, this investigation
could complement nicely the findings presented here.
Response: We appreciate this comment. We agree that meteorological data can provide very
interesting findings, however we would like to focus mainly on the sources of PM$_{10}$ and its
oxidative potential. We have added a statement in section 3.5 to clarify that meteorological
influence cannot be ruled out in the trends observed in this study. Please see a detailed answer
for the comment on lines 371- 382 and a corresponding improvement in section 3.5. We also
provided an answer to a similar question by reviewer 2. We would also like to point out that, in
fact, the STL deconvolution used in section 3.5 (methodology explained in section 2.6)
decomposes the PM10 time series into three components: trend, season, and residual. With a
free amplitude of the seasonal change, this method somehow takes into account the changes in
seasonal cycles from year to year which could also delineate part of the effect of meteorology
on the long-term trend of PM10. This is now included in section 2.6, with the sentence :
The STL (Season-trend deconvolution using locally estimated scatterplot smoothing) model
decomposes the PM10 time series into three components: trend, season, and residual. With a
free amplitude of the seasonal change, this method somehow takes into account the changes in
seasonal cycles from year to year which could also delineate part of the effect of meteorology on the long-term trend of PM10.

Referee comment: Line 15: A specification of the name/location of the site could be given in
addition.

Response: Thank you for the suggestion. We revised this sentence as:

In this study, a 9-year sampling of $PM_{10}$ (particles with an aerodynamic diameter below 10 μm)
was performed in a rural background site in France (Observatoire Pérenne de l'Environnement
or OPE) from February 28, 2012 to December 22, 2020.

Referee comment: Lines 19-22: The difference between the two sentences is not straightforward
and clear. Could you please rephrase and make the difference clearer?

Response: To address the referee's comment, we improved the sentence as:

The sources of OP were also estimated using multiple linear regression (MLR) analysis. In
terms of mass contribution, the dominant sources are secondary aerosols (nitrate- and sulphate-
rich) associated with long-range transport (LRT).

Referee comment: Line 24: But this is not a result of this study, since you did not analyse urban
areas.

Response: We appreciate the feedback. The authors would like to note that all OP
measurements, including those in Figure 7, were analysed by our group at Institut des
Géosciences de l'Environnement (University Grenoble Alpes) and published in different
publications (Borlaza et al., 2021; Weber et al., 2021). References were added to make it clearer,
and we feel that with this, the comparison and discussion of OPE results compared to OP values
from other sites are therefore fully a result from this publication, and should as such be included
in the main text.

Referee comment: Line 29: This sentence is not clear: revise.

Response: We appreciate the feedback. We revised the sentence as:

Particulate matter (PM) pollution is a key factor in various environmental concerns affecting
public health and climate.

Referee comment: Line 35: "geochemical" may be not the most appropriate term in this context.
In addition, the pollutants are transported, not the sources. Please revise.

Response: Thank you for the suggestions, we improved the sentence as:

Rural sites are of great interest as well because they can represent the regional background of
the atmosphere and potential influence from long-range transport (LRT) of pollutants.

Referee comment: Lines 36-37: Well, not only large-scale processes, but also mesoscale processes are needed for chemical transport models. Revise.

Response: Thank you for the suggestion, we improved the sentence as:

Studies at such sites enable the understanding of large-scale and mesoscale processes
(Anenberg et al., 2010; Mues et al., 2013; Konovalov et al., 2009), which is necessary to
elaborate chemical transport models.

Referee comment: Line 38: Again, "geochemical" is not the appropriate term in this context.

Response: Thank you for the suggestion, we improved the sentence as:

The continuing observations in background sites can lead to the identification of long-term
trends and the effect of recent changes in the source emissions.

Referee comment: Lines 55-60: I would suggest presenting the structure of the work rather than
summarizing the results and conclusions.

Response: We appreciate the feedback. To address the referee's comment, we improved Lines
55-60 as:

The OPE site (Observatoire Pérenne de l'Environnement) is located in a rural site in
Houdelaincourt, north-eastern France, well-representing the French national background PM.
The long-term monitoring of $PM_{10}$ (particles with diameter $\leq 10$ µm) over a 9-year period ($n =$
434) in the OPE site allowed an extensive characterization of the chemical and OP of $PM_{10}$.
The objectives of this work are, first, to achieve for the very first time a study of the main
sources of PM in a rural environment in Europe, using a long-term database including several
specific organic tracers in the carbonaceous fraction. The PMF methodology includes a unique
validation with comparison of the chemical profiles of the factors with those obtained in many
other studies in France. The second objective is to quantify the temporal evolution of the
contributions of these sources over the period of the study, particularly focusing for the first
time on the vehicular emission that have already been shown to decrease in urban environments
in Europe during the last decades. Finally, another major objective is to perform the
deconvolution of the contribution of the PM sources to the OP measured with AA and DDT
assays, and to determine the most important sources for the oxidizing capabilities of PM
influencing human health in such an environment.

Referee comment: Line 63: The acronym was introduced previously without explanation: better
move this explanation to the first time it is cited.

Response: The acronym is now defined in Line 55:

The OPE site (Observatoire Pérenne de l'Environnement) is located in a rural site in
Houdelaincourt, north-eastern France, well-representing the French national background PM.

Referee comment: Lines 63-68: Any additional description of the typical local climate? This can affect PM concentrations and may be relevant for the rest of the discussion.
Response: Thank you for this suggestion. We added some information about the local climate
in section 2.1 and reads as:
The mean annual temperature between 2011 and 2018 in the area was 10.5°C [minimum,
maximum: -15.2°C, 36.4°C], average cumulated yearly precipitation was 829 mm, and the
predominant local wind regimes are south-westerly and east-north-easterly winds (Conil et al.,
2019).
Referee comment: Lines 75-78: This means that you analysed only field blanks and not blank
filters? With which frequency did you analyse these field blanks?
Response: Field blank filters (filters subjected to all the same steps of preparation and sampler
loading, but not exposed to sample flow) were collected and analysed regularly, with about 2
to 3 field blanks per month. This amounting to 15% field blanks compared to real samples,
which is a high standard for field collection.
Referee comment: Line 89: What is this "range of ratio"? Please explain better.
Response: Apologies for this typographical error. We clarified this by revising the sentence as:
Yazdani et al. (2021) showed that this is consistent with the range estimated for rural samples
from the IMPROVE network, that are generally higher than for urban samples.
Referee comment: Lines 83-111: Any specifications of the Limits of Detection, and other
experimental parameters?
Response: We appreciate the suggestion. The authors added specifications on the quantification
limits (QL) of each chemical specie measured in the OPE site, please see Table S1 and a
sentence in section 2.2 that reads as:
A summary of the quantification limits (QL) on each chemical specie measured in the OPE site
is also provided in Table S1.
Referee comment: Lines 115-116: If the analysis was started on samples collected from June
13, 2017 to December 22, 2020, it means that then you analysed also the rest of the samples. Is
this true? If not, please revise.
Response: Thank you for the suggestion, we appreciate it. To clarify the statement, the authors
revised the sentence as:
The OP analysis only started on samples collected from June 13, 2017 to December 22, 2020,
amounting to a total of 191 samples.
Referee comment: Line 139: Change "PMF 5.0" to "EPA PMF 5.0".
Response: To address the referee's comment, we improved the sentence as:
The United States Environmental Protection Agency Positive Matrix Factorization (EPA PMF

5.0) software (Norris et al., 2014) was used to identify and quantify the major sources of $PM_{10}$.
Referee comment: Line 140: This definition is not correct: please revise.
Response: We appreciate the feedback. We have revised the definition as:
PMF is a receptor model fully described by Paatero and Tapper (1994) and is now widely used
for source apportionment around the world.
Referee comment: Lines 147-148: Which paper did you follow for this step? And do you know
that this is not complete to characterize the strength of the variables? The analysis of the
residuals should be also made. In addition, how did you treat the additional uncertainty? Please
revise.
Response: Thank you for the feedback. In the supplementary information (S2), the authors have
provided a detailed PMF methodology. The uncertainties were estimated following the method
proposed by Gianini et al. (2012). To make it clearer, the authors improved section 2.4.2 as
follows:
Solutions with a total number of factors from 6 to 11 were tested for the baseline models.
Following the recommendations of the European guide on air pollution source apportionment
with receptor models (Belis, 2019), the $Q/Q_{exp}$ ratio (<1.5), the geochemical interpretation of
the factors, the weighted residual distribution, and the total reconstructed mass were evaluated
during factor selection.
Gianini, M., Fischer, A., Gehrig, R., Ulrich, A., Wichser, A., Piot, C., Besombes, J.-L., and
Hueglin, C.: Comparative Source Apportionment of $PM_{10}$ in Switzerland for 2008/2009 and
1998/1999 by Positive Matrix Factorisation, Atmos. Environ., 54, 149–
158, https://doi.org/10.1016/j.atmosenv.2012.02.036, 2012.
Referee comment: Lines 150-152: How did you analyse weighted residuals? Please provide
additional details.
Response: Please refer to the supplementary information (S2) in Eq. S4.
Referee comment: Lines 164-171: Did you use any particular software for this calculation?
Response: The equations used for the PD-SID metric are mentioned in the supplementary
information (S2) closely following a previous work by our group (Weber et al., 2019, 2021)
and were calculated using Python. See also the answer to reviewer 3 on a similar question.
Referee comment: Line 204: What does it mean "reconstructed"? How far is this reconstruction
from the measured value?
Response: Thank you for the clarification. Reconstructed PM mass is the mass calculated from
chemical characterization (i.e. total PM reconstructed from all chemical analysis performed).
We have improved Figure 2 to estimate the amount of unknown species (e.g. the semi volatile
fraction) in the total measured $PM_{10}$.

Referee comment: Lines 211-212: A decrease during this period is quite evident, while the
change in composition is less evident.
Response: Apologies for the confusion, we have revised the sentence as:
Some changes in the concentration may arise in the PM10 mass concentration, but changes in
the major chemical components at the OPE site are less visible, even with the lockdown
restrictions during year 2020.
Referee comment: Line 215: Is this a mean value? From Figure 2 this value does not seem
constant.
Response: Yes, this is the mean percentage contribution of OM to total $PM_{10}$. To clarify, the
sentence was improved as:
Accounting for 37% to 45% (based on year) of the reconstructed $PM_{10}$ mass concentrations,
organic matter (OM) is the largest contributor.
Referee comment: Lines 219-220: This example does not explain much. Please revise.
Response: We appreciate the feedback. However, the example serves as a general comment
about expected species in vehicular emissions and road dust. These were further clarified in
section 3.3, where both factors were discussed in detail.
Referee comment: Lines 220-221: This sentence is not well linked with the previous results
shown. Please revise.
Response: Thank you for the feedback. The authors deem that the sentence could be useful in
linking the composition of PM (section 3.1) and the sources identified by PMF (section 3.2 and
3.3), and then consequently should be an essential step for efficient air quality policies.
Referee comment: Lines 223-225: Please provide additional details on how this solution was
selected. The signs of instability are worrying.
Response: We appreciate the feedback. In section 2.4.2 and 2.4.3, the criteria for a valid solution
and the appropriate constraints in the PMF model were discussed. Additionally, the authors
have allotted section 3.2 to discuss the statistical stability of the PMF solution. Finally, Figures
S2 to S10 presents the chemical profile and temporal evolution with error estimates of the PMF-
resolved factors. Please refer to the improved version of section 2.4.2.
Referee comment: Lines 233-235: Couldn't you use a value more appropriate for a rural site?
Response: We have tried a range of values, but none has led to better PMF solutions. Hence,
the decision of not applying such constraints.
Referee comment: Lines 263-265: But local sources cannot be excluded: if not, we would have
always higher nitrate and sulphate concentrations at rural sites than at urban ones, which is not

388 the case.

390 Response: The authors agree with the referee that local sources should not be excluded.
391 However, the OPE site is located in a remote area in the north-eastern part of France (48.5°N,
392 5.5°E) and without any residential areas within several kilometres (see Figure 1). The density
393 is about or less than 5 inhabitants per $km^2$ within a radius of 10 km around the site. The sentence
394 in Line 263 to 265 mentioned that sulphates and nitrates are mainly formed through secondary
395 processes with long atmospheric lifetimes and can originate from regional sources or LRT.
396 With the *a priori* knowledge of the site description, the authors deemed it was appropriate.

398 Referee comment: Line 270-271: Can you explain the reasons of this seasonality in these 2
399 factors?

401 Response: In France, biomass burning emissions are expected to be more prominent in the
402 winter due to residential wood burning, while the seasonality in the nitrate-rich factor can due
403 to greater photochemical activities in spring associated with agricultural spreading of manure
404 and fertilizers. The general characteristics of the PM over France are presented in Favez et al.,
405 2021).

407 Referee comment: Lines 296-297: Couldn't this be due to the fact that Na+ and Mg2+ are
408 primary seasalt particles while MSA particles are secondary? What about the correlation with
409 nss-SO42-? See for instance papers from the group of Silvia Becagli and Roberto Udisti (e.g.,
410 Udisti et al., 2016; Becagli et al., 2019, 2021). Did you try to analyse the source (with wind or
411 back-trajectories)?

413 Response: We appreciate the comment. The OPE site is 340 km away from the closest sea,
414 making it difficult to assume that these are, essentially, primary sea salt particles. Road salting
415 in the winter could potentially be an origin of salt particles. The discussion on seasonal
416 variability and sources of MSA (together with a PSCF analysis) in the OPE site (together with
417 4 other rural sites in France) can be found in Golly et al., 2019 (already in the list of references,
418 doi.org/10.1016/j.atmosenv.2018.10.027). Our group is also currently working on a synthesis
419 paper discussing MSA concentrations and its sources across France, using about 20 yearly
420 sampling campaigns from different sites, in order to discuss the links with marine source and
421 transport.
422 The mentioned articles by Silvia Becagli and Roberto Udisti were studies performed in marine
423 areas in the Arctic or Antarctic, which is not totally relevant for our conditions. However,
424 another publication by Becagli (http://dx.doi.org/10.3189/172756405781813384) says: "The
425 spatial distribution of nssSO42− and MSA is discussed as a function of distance from the sea,
426 altitude and accumulation rate. Depositional fluxes of nssSO42− and MSA decrease as a
427 function of distance from the sea, with a higher gradient in the first 200 km step." It follows
428 that concentrations further away from coastline should be quite low.

430 Referee comment: Figure 6: Check the y-scale (measurement unit).

432 Response: To clarify the measurement unit, the figure caption was revised as follows:

434 Figure 6: The Season-trend (STL) deconvolution of contributions in µg m$^{-3}$ from the traffic
435 factor to PM$_{10}$ from year 2012 to 2020.

437 Referee comment: Lines 371-382: There are also studies at high-altitude or regional background sites, some of which highlighted a concurrent role of a changing meteorology and of a change
in the frequency of Saharan dust advections to Europe. Please look at: Tsyro et al., 2018; Colette
et al., 2011, 2017; Brattich et al., 2012, 2020 and references therein). Thus this discussion may
be improved.
Response: We appreciate the suggestion. We can remark that most of the literature quote is
from modelling work, with all the questioning related to the capabilities of models to clearly
reproduce the local concentrations in the boundary layer. We note also that Collette et al. (2011)
clearly state that the decreasing emissions during the last decades largely dominate any impact
of the meteorological conditions for species of anthropogenic origins. The work of Brattich et
al. (2020) is quite interesting with a thorough comparison of regional circulation patterns with
local measurements, but is performed for Mont Cimone, an altitude site in the free troposphere
where the large-scale circulation is much more important than the local (a few hundred km)
one. For example, average winter PM10 concentration of less than 3-4 μg m$^{-3}$ at this site cannot
be said representative of boundary layer rural concentrations in Europe, but much more related
to large scale transport only.
However, we have improved this section and now reads as:
The downward trends found in our study are well consistent with other existing studies in
Europe (Li et al., 2018; Sun et al., 2020; Salvador et al., 2012; Pandolfi et al., 2016; Gama et
al., 2018; Amato et al., 2014), nearly all of them conducted in urban areas. Pandolfi et al. (2016)
found a significant long-term decrease of the contributions from anthropogenic emissions
(specifically a mixed industrial/traffic factor, -0.11 μg m$^{-3}$ year$^{-1}$, 56% total reduction) in a
regional background site in altitude in northeast of Spain (Montseny, Spain) from 2004 to 2014.
This is also consistent with a similar study in the metropolitan area of Madrid, Spain (Salvador
et al., 2012) which showed a reduction of 32.7% attributed to traffic emissions, alongside the
decrease of the carbonaceous and SO$_4^{2-}$ in PM. In a southern Spain area (Andalusia), the same
group also found a consistent decreasing trend of PM at some traffic and urban sites in the
region (Amato et al., 2014).
Another long-term study in Central Europe (Sun et al., 2020) focusing on eBC concentrations
found decreasing trends in high-altitude Alpine sites located in Germany (-3.88% year$^{-1}$, [-
10.15%, 0.56%]) and Switzerland (-3.36% year$^{-1}$, [-8.71%, -0.28%]). These findings are also
consistent with results from other parts of Europe, with the largest decrease found in OC up to
-48% (Cusack et al., 2012) and the decrease in PM has been associated to non-meteorological
factors (Barmpadimos et al., 2012). Other studies with pluri-annual series of data on PM
chemistry in rural environments in Europe includes Splindler et al. (2013) (Melpitz, Germany,
including EC measurement for 2003-2011), and Grange et al., (2021) (Payern, Switzerland
comparison of 3 periods every ten years since 1998, including EC and trace elements). Both are
showing decrease of EC concentrations over time during the study. Finally, while these studies
did not target specific chemical species solely linked to vehicular emissions, most of them
attributed the decline to the efforts to reduce vehicular emissions and other mitigation policies
in their respective areas.
It should be noted that the role of meteorology on the observed decrease in PM in these studies
(including ours) cannot be totally ruled out (Hou and Wu, 2016; Czernecki et al., 2017; Kim,
2019) and is generally not fully considered. However, the complex interplay of all
meteorological variables on PM concentrations would be difficult to delineate. Indeed, there
are some studies at high-altitude or regional background sites that highlighted a concurrent role
of changing meteorology and changes in frequency of Saharan dust advections to Europe
(Brattich et al., 2020) in modulating the dust concentrations in the atmosphere. The study at

Melpitz (Spindler et al., 2013), despite an in-depth work on the wind sector classification, does
not address the impact of possible changing in the air mass origin on long-term changing
origins.
Referee comment: Figure 8: Please check the x-axis and add more ticks. There are some peaks
in all series: did you analyse the presence of outliers and investigate their causes?
Response: Thank you for the suggestion. Figure 8 was updated accordingly. There are very few
data points ($n$=3) where the $PM_{10}$ mass concentration was higher than usual. These samples did
not exhibit particularly high OP activities. However, we have observed elevated levels of
contributions from the nitrate-rich factor for about 9 to 12 times as much as the over-all average
contribution.
Referee comment: Lines 511-514: This is not clear: the problem with weekly samples should
be that differences and transient events (e.g., Saharan dust, fires, …) are smoothed but I cannot
understand what you mean by "the weekly collected samples may contain particles that are not
fully captured in a daily sample".
Response: We appreciate the comment. Weekly samples contain both weekend and weekday
samples as the duration of collection is 7 days. On the other hand, a daily sample can either fall
on a weekday or a weekend as samples are collected on a 6-day interval. This implies that the
weekly collected samples may contain particles that do not have the same representability than
daily samples. The authors hope that this clarifies the confusion.
Referee comment: Line 530: Figure 11 is not about PM concentrations, but on OP contributions.
A decrease in PM concentration could instead be observed as previously noted.
Response: Figure 11 is part of the section that discuss the sources of OP of $PM_{10}$. The main
take-away of the figure is that the available OP data only spans up to 4 years against 9 years for
the PMF data. The shorter time range could be insufficient to reach significance and robustness
in the trend assessment of OP levels compared to the trends analysis we have performed for
PMF (Figure 6).
Referee comment: Lines 548-550: As previously noted, this discussion is limited since there
are studies evidencing a simultaneous effect of the changing meteorology. This point should be
improved.
Response: This paragraph has been improved and now reads as:
Thanks to the long-term dataset in the OPE site, it was observed that the traffic factor
contribution to total $PM_{10}$ has decreased over the years for this site that may well represent the
French national background PM. This decrease is much larger than any change observed for
the other PM sources and is in excellent agreement with estimations in the decrease in BC
emissions from the transport sector all over France from the national inventory. This effect may
be attributed to improvement of the exhaust emission of terrestrial transportation fleet, and/or
to regulations restricting vehicular emissions in bigger cities and/or other regional-scale.
However, persistent changes during the same period in some meteorological processes
influencing the transport of air masses to OPE or formation of PM during this transport cannot
be totally ruled out. We would also like to point out that, in fact, the STL deconvolution used
in section 3.5 (methodology explained in section 2.6) decomposes the PM10 time series into three components: trend, season, and residual. With a free amplitude of the seasonal change, this method somehow takes into account the changes in seasonal cycles from year to year which could also delineate part of the effect of meteorology on the long-term trend of PM10. This is now included in section 2.6, with the sentence :

The STL (Season-trend deconvolution using locally estimated scatterplot smoothing) model decomposes the PM10 time series into three components: trend, season, and residual. With a free amplitude of the seasonal change, this method somehow takes into account the changes in seasonal cycles from year to year which could also delineate part of the effect of meteorology on the long-term trend of PM10.

Referee comment: Code and data availability: Please check the statement for this: as on the ACP website https://www.atmospheric-chemistry-and-physics.net/policies/data_policy.html, "Authors are required to provide a statement on how their underlying research data can be accessed. … If the data are not publicly accessible, a detailed explanation of why this is the case is required. … Data do not comprise the only information which is important in the context of reproducibility. Therefore, Copernicus Publications encourages authors to also deposit software, algorithms, model code, video supplements, video abstracts, International Geo Sample Numbers, and other underlying material on suitable FAIR-aligned repositories/archives whenever possible"

Response: Thank you for this advice. We appreciate the reminder.

**Response to anonymous referee #2:**

Referee comment: The samples were collected in a 6-day sampling interval, but not weekly samples covering each year. The weakness should be discussed if any real-time measurements of PM10 are available

Response: Thank you for the suggestion. The authors would like to point out that a sampling interval of 1 every 3 days or 1 every 6 days has been reported as appropriate for meeting general monitoring objectives. An intensive sampling interval, however, is recommended for areas expected to exhibit higher ambient levels (Bortnick et al., 2002). The OPE site, being a rural background site, does not exhibit levels that require an intensive sampling interval.

Referee comment: Meteorological conditions would affect the long-term trends, and the issue should be quantified. This is particularly critical for semi-volatile species such as ammonium nitrate and organics when their interannual variations were analyzed.

Response: We appreciate the insight. We agree that meteorological conditions can have a wide range of impacts on PM concentrations. However, this is a really intricate issue, since both temperature, humidity, amount of rain, amount of radiation, wind, air mass origin, boundary layer height, amongst others can be at play, both locally but also during the few days during the transport to the site. Investigating the evolution over the last 10 years of all these parameters and the potential impact on PM is altogether a really large task, and most probably beyond the current state of the art. The authors are not sure that there is a single published study that investigated all these parameters (plus the PM sources of mass and OP) when studying a time series of observations.

We have added a statement to clarify that meteorological influence cannot be ruled out in the trends observed in this study. Please refer to answers to reviewer #1's comments and associated changes in the text. We would also like to point out that, in fact, the STL deconvolution used in section 3.5 (methodology explained in section 2.6) decomposes the PM10 time series into three components: trend, season, and residual. With a free amplitude of the seasonal change, this method somehow takes into account the changes in seasonal cycles from year to year which could also delineate part of the effect of meteorology on the long-term trend of PM10. This is now included in section 2.6, with the sentence :

The STL (Season-trend deconvolution using locally estimated scatterplot smoothing) model decomposes the PM10 time series into three components: trend, season, and residual. With a free amplitude of the seasonal change, this method somehow takes into account the changes in seasonal cycles from year to year which could also delineate part of the effect of meteorology on the long-term trend of PM10.

Referee comment: Considering agriculture and natural emissions of ammonia, secondary aerosols could be formed before and during the long-range transport as well as formed locally. This may should be clarified.

Response: We appreciate this comment. We have incorporated this in the manuscript, see section 3.3. We have added a line that reads as:

Considering the agriculture and natural emissions of ammonia, especially expected in a rural site, secondary aerosols could also be formed locally in the OPE site.

Referee comment: Nacl could be derived from road salts or sea salts considering an inland site used in this study, please clarify.

Response: We appreciate this comment. We have incorporated this in the manuscript, see section 3.3. The paragraphs read as:

The **aged sea salt** factor is characterised by high loadings of $Na^+$ and $Mg^{2+}$, with a certain amount of species originating from potentially anthropogenic sources such as nitrates (6% of $NO_3^-$ mass) and sulphates (19% of $SO_4^{2-}$ mass) that can be attributed to mixing and transformation processes in the atmosphere. Interestingly, there are some contributions from EC (8% of EC mass), Cu (11% of Cu mass), Sb (13% of Sb mass), and Se (19% of Se mass). This could imply potential mixing of aged sea salt with other anthropogenic source linked to these species (e.g., traffic, shipping). The minimal loadings observed in the contributions of $Cl^-$ in this factor can be a likely result of ageing processes occurring between sea salt and acidic particulate compounds such as nitric and sulfuric acid (Seinfeld and Pandis, 2016). This factor could also be associated to road salting in the winter, however there is no clear seasonality in the contributions to support this hypothesis. There was no added constraint in this factor as our solution shows a $Mg^{2+}$ to $Na^+$ ratio at 0.06 while this ratio is usually found around 0.12 in sea-salt emissions (Henderson and Henderson, 2010).
The **fresh sea salt** factor is characterised by high loadings of $Cl^-$ (91% of $Cl^-$ mass) and some contributions from $Na^+$ (35% of $Na^+$ mass) and $Mg^{2+}$ (25% of $Mg^{2+}$ mass). This factor contributes 4% to total $PM_{10}$ mass and, unlike the aged sea salt factor, it is less likely influenced by anthropogenic sources with extremely low contributions from carbonaceous and metal species.

Referee comment: Lines 276-277, "The NH4+/NO3- mass ratio in this factor is 0.22, close to the mass ratio (0.29) indicating the formation of NO3NH4 in the particulate phase." The statement does not sound scientific considering PM10 to be studies; the same comment is applied to lines 281-282" The NH4 + to SO4 2- molar ratio of 0.4 suggests that sulphates are mostly present as (NH4)HSO4 and only a small fraction as (NH4)2SO4.

Response: We removed these sentences, since they are not essential for the discussion of the paper.

**Response to anonymous referee #3:**

Referee comment: Many of the results related to the PM10 analysis are neither surprising nor original, i.e. traffic-related concentrations are decreasing (even in rural areas), SIA are large contributors to the total mass of PM10 in rural areas. What did we learn from your analysis that has general implications for atmospheric science, and was previously not known? Citing from the scope of ACP, "The journal scope is focused on studies with general implications for atmospheric science rather than investigations that are primarily of local or technical interest.". In other words, I feel that a clear scientific hypothesis/research question is lacking in this manuscript, and perhaps the authors should elaborate more on what they are trying to uncover. The process of a scientific manuscript in a high-quality scientific journal (as ACP) should start with a clear question/hypothesis that authors should try to answer given some new observational data/new modeling experiments/new theoretical insights. Reading the manuscript, I felt that the focus of the paper is on the dataset itself, rather than using the dataset as a tool to answer a scientific question. As an example of this, the results from Pandolfi et al. (2016) also suggest that traffic concentrations in a rural background site (in Spain) have decreased in the last few years and secondary inorganics and organics are large contributors to PM10 in rural sites. For the PM10 results, what did we learn from your analysis that adds significant new knowledge with respect to Pandolfi et al. (2016) work (to cite one, but there are other similar works available in the literature)? This should exceed the simple time span differences considered in their work. With that said, I do acknowledge that there is some merit in analyzing OP contributions and show the differences with PM10 contributions (Sections 3.6-3.8), as OP is emerging as a promising endpoint-related metric to measure health impacts. However, in the introduction, you state that 'The characterization of PM sources and OP in a rural site will enable us to see the large-scale effects of mitigation policies that target reduction of PM mass concentrations. This will also provide knowledge of efficiency of current air quality guidelines in terms of other emerging health-based metrics of PM exposure.' (Lines 51-54). A similar statement is repeated at the end of the introduction (lines 58-60). What knowledge of efficiency of current air quality guidelines for OP did you find? This is not clearly reflected neither in the results nor in the conclusions, as the trends analysis for OP is not really revealing much given the relatively short (4 years) period available. Once again, I believe you should try to highlight the value of your analysis as it relates to a specific hypothesis/questions, rather than propose very general statements that confuse the reader.

Response: We really want to thank the reviewer for this strong comment that led us to think more about our work, and to read in more detail the work by Pandolfi's et al (2016) and some other pieces of work. All of these lead to several modifications in this present version of the paper that (according to us) fairly improved it.
Considering Pandolfi's work and what our work adds compared to it, as per referee's demand, we had a much deeper look at this seminal and interesting study conducted a while ago by excellent colleagues. In the end, we found out that our work is quite different from this paper, but also that the conclusions in Pandolfi et al. (2016) are not as strong as presented by the reviewer. We think that some (we hope significant) aspects are somewhat innovative with our work, compared to Pandolfi's one, but also compared to all other papers dealing with long time series of sources of PM in rural environments in Europe.

- Our site is in a location that probably reflects more rural areas in Europe where people are living, compared to Montseny which is in altitude.
- Our work is including several innovative sources due to the inclusion of organic tracers that were not delineated in Pandolfi's work, nor in any previous study: biomass burning, MSA-rich, primary biogenic; the domestic biomass burning source is really important to evaluate for regulation purposes. Altogether, these "new sources" represent 24 % on average of the $PM_{10}$ mass, and a much larger share of the organic matter mass. They were not investigated together previously in any published paper concerning long term trends in a rural area in Europe, and in itself, this makes our work innovative on a scientific point of view.
- The time series at Montseny in Pandolfi's work do not allow to detect any trends for EC nor OC. This is a clear sign that the source from traffic is, indeed, not likely to present a significant decrease, as observed from this site. The time series of EC at our site present a really clear decreasing trend, the most important of all chemical species measured. This is a huge difference.
- The PMF factor labelled industrial / traffic for Montseny in Pandolfi's the main text of the paper (and anthropogenic in its SI) does not include EC as a tracer but a global carbonaceous indicator (non-mineral C). The authors present this factor like much more influenced by industry than by traffic. Hence, EC is not decreasing, and there is no clear sign that a clearly identified traffic source is decreasing.
- The OP of PM has been gaining attention as an emerging health-based metric of PM exposure. The findings in this study, particularly the identified main drivers of OP of PM, could help validate the relative importance of some PM sources in terms of adverse health effects. For example, the nitrate-rich source is the highest contributor to PM mass but has minimal contributions in terms of OP (Figure 10 in the manuscript). So, setting a regulation targeting mass concentration during elevated ammonium nitrate events may have less adverse impacts on health than expected.

There are, of course, several other differences that makes our work innovative (according to us) on a scientific point of view, including the STL deconvolution that allows to precisely investigate tendencies, and of course the all section on OP measurements, as underlined by the reviewer. Even if we agree that 4 years of OP analysis is a bit short for trends analysis, we have not identified much studies that have published more than one or two years of OP time-series and, usually such publications rely heavily on reconstructed data (Fang et al. (2014), doi:10.5194/acp-14-12915-2014).

The comment by the reviewer led us to state the objectives of this work in a more explicit way at the end of the introduction, and we hope that it is now clearer to highlight the motivation of the study. This now reads as:

The OPE site (Observatoire Pérenne de l'Environnement) is located in a rural site in Houdelaincourt, north-eastern France, well-representing the French national background PM. The long-term monitoring of $PM_{10}$ (particles with diameter $\leq 10$ μm) over a 9-year period ($n = 434$) in the OPE site allowed an extensive characterization of the chemical and OP of $PM_{10}$. The objectives of this work are, first, to achieve for the very first time a study of the main sources of PM in a low altitude rural environment in Europe, using a long-term database including several specific organic tracers in the carbonaceous fraction. The PMF methodology includes a validation step with the comparison of the chemical profiles of the factors with those
obtained previously in many other studies in France. The second objective is to quantify the
temporal evolution of the contributions of these sources over the period of the study,
particularly focusing for the first time on the vehicular emission that have already been shown
to decrease in urban environments in Europe during the last decades. Finally, another major
objective is to perform the deconvolution of the contribution of the PM sources to the OP
measured with AA and DDT assays, and to determine the more important sources for the
oxidizing capabilities of the PM influencing human health.

Referee comment: In section 2.3, the difference between the two assays can be elaborated
further. Are the large differences presented in Figure 9b and Figure 9c expected? Are the
contributions to total OP completely different because the two assays are meant to look at
different oxidative processes in the lungs? This can be discussed in more details either in the
methodology (highlighting more clearly why the two assays are used) or in the results, when
commenting on the differences between Figures 9b and 9c.

Response: Thank you for this comment. To address the referee's comment, we improved
section 2.3 as follows:

DTT is used as a chemical surrogate to mimic *in vivo* interaction of PM with biological reducing
agents, such as adenine dinucleotide (NADH) and nicotinamide adenine dinucleotide phosphate
(NADPH), in the DTT assay. The consumption of DTT in the assay represents the ability of
PM to generate ROS (i.e., superoxide radical formation) (Cho et al., 2005). The $PM_{10}$ extract is
mixed with the DTT solution. Afterwards, the remaining DTT that did not react with $PM_{10}$ is
titrated by 5,50-dithiobis-(2-nitrobenzoic acid) (DTNB). This reaction produces 5-mercapto-2-
nitrobenzoic acid or TNB. The TNB is measured by absorbance at 412 nm wavelength using a
plate-reader (TECAN spectrophotometer Infinite M200 Pro) with 96-well plates (CELLSTAR,
Greiner-Bio) in a 10-minute time step interval for a total of 30 minutes of analysis time.
AA is a known lung antioxidant used in AA assays using a respiratory tract lining fluid (RTFL)
(Kelly and Mudway, 2003). This antioxidant prevents the oxidation of lipids and proteins in the
lung lining fluid (Valko et al., 2005). The consumption of AA also represents PM-induced
depletion of a chemical proxy (i.e. cellular AA antioxidant). The mixture ($PM_{10}$ extracts reacted
with AA) is injected into a 96-well multiwall plate UV-transparent (CELLSTAR, Greiner-Bio)
and measured at 265 nm absorbance using a plate-reader (TECAN spectrophotometer Infinite
M200 Pro) in a 4-minute time step interval for a total of 30 minutes of analysis time.
Both DTT and AA assays measure OP by depletion of specific chemical proxies, cellular
reductants (for DTT) and antioxidants (for AA). Studies have well-identified a large number of
PM constituents that influence OP concentrations. At least, OP assays are known to be
associated with some metals (Cu, Fe, Mn among others) and some organic species (especially
photochemically sensitive species such as quinones) (Calas et al., 2017, 2019, Charrier et al.,
2014, Pietrogrande et al., 2019). However, in ambient air, each assay reports its own
associations that may vary according to the local context (emission sources, local transport
leading to various ageing processes and spatiotemporal variations) (Gao et al., 2020). Hence,
a synergetic approach using multiple OP assays, to capture the most complete information
regarding PM reactivity, is commonly suggested. (Bates et al., 2019, Calas et al., 2017, Borlaza
et al., 2021)).

Referee comment: I have some concerns about the MLR method presented in Section 2.5. One
of the key assumption of linear regression is that the residuals ε are iid, but you're using the

MLR to model timeseries data, where the assumption is evidently violated (by definition the
OP data are not independent, as there is a clear temporal dependence). Perhaps you should
consider adding a temporal component to your model (e.g. ARMA) that takes care of the
temporal dependence to avoid misinterpreting the results on the $\beta$ coefficients. Adding a
temporal component is somewhat equivalent to detrend the data and remove seasonality, as you
seem to be doing for PM10 analyses but not for the MLR part.

Response: We appreciate the referee's feedback. The goal of MLR is to apportion observed OP
to PMF-resolved sources allowing to determine the main drivers of OP. The $\beta$ coefficients
(intrinsic OP) in the MLR model represents the OP property of each $PM_{10}$ source. The source-
specific OP contribution is calculated by multiplying the intrinsic OP of each source by the
mass contribution of the source to total $PM_{10}$. ARMA is a great model for forecasting and can
be used both for seasonal and non-seasonal time series data, however forecasting is not the
purpose of MLR in this study.

Referee comment: In the same section, I am not sure if I am interpreting Equation 1 correctly.
Why is there a subscript to your and $\beta$ matrices? In previous sections, referred to the total
number of observations whereas the meaning of here is unclear. Shouldn't your model
simply be:
$OP = G\beta + \varepsilon$
With my comment above, the model should be extended to something like:
$OP(t) = \alpha OP(t-1) + G\beta + \varepsilon$
To take care of the temporal dependence. Obviously the choice of AR(1) is very simple but
different AR or ARMA models can be investigated. Also if G is a matrix, what does it mean
the subscript ? I believe you should double check your notation to be consistent. If you decide
to use matrix notation, you should be consistent throughout.

Response: Thank you for the clarification. Apologies for the confusion. The G matrix is the
PMF-resolved source contributions and the subscript denotes the specific source (i.e., biomass
burning, traffic, nitrate-rich, etc.). $\beta$ is the regression coefficient representative of the intrinsic
OP of each source as well.

Referee comment: At the end of the introduction (line 59), you mention long-term trends of
emission sources. Please be sure to use the right words here, I do not believe you are discussing
emission trends at all, but only the decomposition of PM10 concentrations in different PMF
factors.

Response: We appreciate the feedback. The sentence was revised and now reads as:

Finally, another major objective is to perform the deconvolution of the contribution of the PM
sources to the OP measured with AA and DDT assays, and to determine the more important
sources for the oxidizing capabilities of the PM influencing human health in such an
environment.

Referee comment: wonder if an effort could be made to actually show these emission data
(aggregated at some level, for instance for the traffic category in a certain region), and see if
there is some sort of correlation with the concentrations of the traffic factor that you showed
from your analysis). A recurring theme of your manuscript is about analyzing the effect of
recent changes in the source emissions (e.g. line 38-39), but I actually have not seen emission
data at all. Can you make an effort to better substantiate that the reduced concentrations from the traffic sector (Figure 6) are actually related to emissions rather than, say, changing
meteorology?
Response: We appreciate the insight, that led us to effectively search for emissions data for
comparison with our results. In fact, the Citepa is the official organization in France in charge
of providing the emissions inventories at the national level. In their database, they have
provided the evolution of black carbon (BC) emissions from 1990 to 2020 by different sectors
(data available in https://www.citepa.org/fr/2021-bc/). The BC emissions by the transport sector
decreased from 30.9 kt in 1990 to 6.8 kt in 2020 and is fully ascribed to the improvement of
motorization and other traffic reductions.

[Figure]

In Figure 7 in the manuscript, we have provided a comparison of the evolution of the traffic
factor source contribution at the OPE site obtained with our PMF study, and the BC emissions
by the transport sector (source: CITEPA, https://www.citepa.org/fr/2021-bc/) for overall
France. We have improved section 3.5 with an additional discussion about this that reads as:
The evolution of the absolute concentration of the traffic factor at the OPE site was also
compared to an evaluation of black carbon (BC) emissions by the transport sector for overall
France, provided by the CITEPA, the official agency in charge of the emissions inventory in
France (https://www.citepa.org/fr/2021-bc/). Both series were converted in percentage change,
using 2012 as the base 100 year (**Erreur ! Source du renvoi introuvable.**). This figure shows
an excellent agreement in the trend and in the total decrease for estimated BC emissions from
traffic (-64%) and the traffic source contributions observed at the OPE site (-52%), between the
years 2012 to 2020.

[Figure]

**Figure 1: Comparison of the evolution of the traffic factor source contribution at the OPE site and the black carbon (BC) emissions by the transport sector (source: CITEPA, https://www.citepa.org/fr/2021-bc/) for overall France.**

Referee comment: In Figure 2, it might be helpful to add error bars showing one or two standard deviations of your annual data. This would help interpreting how 'significant' (at least on a visual level) the differences between different years are.

Response: Thank you for the suggestion. However, Figure 2 mainly presents the average contribution of major components of $PM_{10}$. A better visualization of trends (monthly, seasonal, fit tendency) in terms of $PM_{10}$ is presented in Figure S11.

Referee comment: Section 2.4.4 is quite unclear (or at least it's unclear until reading the results related to that section). I believe rephrasing the last couple of lines (166-170) and better defining what is implied by 'homogeneous' and 'heterogeneous' will help the reader in the interpretation of the associated results.

Response: Thank you for this comment. This section was revised as follows:

To investigate further any differences in the chemical profiles at the OPE site compared to those obtained at other French sites, a test of similarity was performed using the Pearson distance (PD) and standardized identity distance (SID) metric. This is calculated using Eq. S5 and Eq. S6 in the SI (S2) (Belis et al., 2015), closely following a previous work by our group (Weber et al., 2019). This comparison is based on the source relative mass composition, which allows the evaluation of the variability of PMF solutions across different sites. In this case, the chemical profiles obtained for the OPE site were compared against 15 different other sites over France. A "homogenous source" tends to have a similar profile over different site types and should have PD<0.4 and SID<1.0 (Pernigotti and Belis, 2018). Conversely, the sources with PD and SD values outside of this range are considered as "heterogeneous sources".

---

## Referee Report (RR1)

**Second revision of "9-year trends of PM10 sources and oxidative potential in a rural background site in France" by Borlaza L.J. et al.**

**General comments**

The authors have replied to most of the comments raised by the reviewers. However, I noticed that in some parts the reviewers expressed similar concerns, and the reply from the authors is not always totally satisfying. This for example applies to the effect of meteorology on the trends investigated. Meteorology have a clear and dominant influence on most of the identified factors, and the signal is not always a seasonal effect as stated by the authors. This is never investigated or mentioned in the paper. There is no effort in studying how for instance transport drives the time pattern of some factors. Also, the STL deconvolution shows which are the main component(s) driving the temporal pattern of PM10 and of the different factors: the authors never try to investigate whether the trend reconstructs or not the most part of the signal, and use this analysis solely for the purpose of identifying the presence of a trend in the time series. The original scope of the STL analysis is to decompose time series splitting time series into trend, seasonal and remainder component, and not to obtain and quantify trends which is the step further and needs other types of analysis. In order to investigate the presence of trends, other methods are more appropriate; conversely, this method is really to quantify how much of the signal is due to each of the three components. While in general I would recommend to provide more references to understand how the presented trends are obtained, I would recommend to make an effort so as to make this analysis more complete going in the direction of what all the reviewers have previously commented. In view of these comments and seeing the reduced efforts to align to the requests of the reviewers, my recommendation is for major revisions of this paper.

**Specific comments**

In many cases, I do not understand the revision in the tracked change version since there is no or very small change with respect to the original version and the deleted sentence is just the same as the newly introduced one.

Lines 34-35: This sentence is not clear: rephrase.

Line 39: and what about aerosol transport apart from its formation?

Line 45: well, it is quite restricting to say that the understanding of such processes is only related with the "elaboration" (perhaps also not the most appropriate term) of chemical transport model.

Lines 47-48: This is a general characteristic of the long term time series, and not just of the ones collected at background sites.

Lines 71-73: This detail is not needed at this point.

Lines 69-77: The scopes are very specific and citing already methodologies which happen to be still rather obscure to the reader. Please try to generalize the objectives leaving the details of the methodology for later on in the text.

Line 140-141: Absolutely not clear how you reconstructed PM10.

Lines 142-144: This detail is not useful here, as the reader does not know anything of this comparison.

Lines 206-207: There are other reasons to increase the uncertainties of some variables (indicating them as weak) or excluding them from the analysis; I assume that this has been considered. also, have other sources of uncertainty (e.g., flow rate) taken into account?

Line 211: How did you take into account the weighted residuals distribution? I mean, could you explain how you analysed it, such as you did with the Q/Qexp ratio?

Lines 265-284: Which software or code did you use to apply this analysis?

Lines 266-268: I cannot understand this: meteorology does not have only a seasonal signal, and also how can the interannual variation in the seasonal signal be connected with the effect of meteorology? If you have references, please provide them, because this justification is not convincing.

Line 298: To be true, I can observe also a reduction in NO3- and NH4, which should be among the main chemical species. So I still cannot understand this.

Lines 299-301: Can you explain at least tentatively the reasons of such differences?

Line 378: A part of sulphates has a marine origin.

Lines 384-386: And what about the sulphates to Na+ ratio? Did you observe if there is a particular wind direction for this factor? Or reasons to suspect collinearity? Or any other investigation on this factor which could be also a mixed source?

Lines 388-390: And what about the ratio of Cl- to Na+? Is there chlorine depletion?

Lines 413-416: And what about your study?

Lines 417-424: Any particular temporal pattern for this factor?

Line 432-434: Also here, any particular temporal pattern?

Lines 458-530: Apart from the analysis of trends, could you explain more how to interpret the results of the STL analysis for example in terms of different importance of the three signal components?

Figure 6: The Figure has poor resolution. Also, there is no unit of measurement on the y-axis.

Line 579: Missing reference to a Figure.

Lines 660-666: This sentence is still not clear, and has to be revised.

Lines 702-703: Do you mean the improvements in the technology?

Lines 704-705: Not clear what you mean by "persistent changes". Revise.

Code and data availability: still not in line with the policy of this journal.

Table S2 and text: the "nitrate-rich" and "sulfate-rich" factors should be "secondary nitrate" and "secondary sulfates"? also, the "fresh sea salt" and "aged sea salt" share much of the fingerprint, so there is no clear evidence of why they should be separated in two factors, at least from this table.

Figure S1 and text: is the temporal variation of biomass burning in agreement with your expectations? Does it increase in winter (it seems so) and with wildfires?

Figure S4: About the mineral dust factor (possibly better named "resuspension", as mineral dust would be just the transport of dust from the desert), I would expect an increase in the summer season with less precipitation, and with Saharan dust transports. Is this the case?

Figure S1-S9: In most cases the BS and DISP bars (in the legend, but are actually lines?) cannot be seen. Also, apart from the temporal pattern, could you investigate the effect of meteorology at least in some of these factors?

---

## Referee Report (RR2)

In the following, my comments are provided in red color, below each comment and author's response.

 1. Our reviewers have been insisting on trying to delineate the "trends that were caused by internal annual variabilities of weather or climate conditions". However, it was assumed already in the initial version that the search for all the causes of the temporal evolutions of source contributions was not in the scope of our work, and it is still not in the direction that we want for this paper. This would be a totally different work, that would require gathering many other data (on local and large-scale meteorology, on emission inventories, on parameters that are influencing atmospheric chemistry for the formation of secondary chemical species, …) on the long term, and it is not our purpose. One aspect of our paper is focused on showing that the data set allows to identify trends in the evolution of the sources contributions of PM for this rural site, and that these trends are different for one source to another. All of this is already an innovative step compared to most of the (few) similar papers in the literature that discuss of evolutions of the concentrations of some chemical species for similar sites, or evolutions of sources with a set of tracers that is much less elaborated than ours. We believe that this innovative result by itself justify the publication, as a step ahead of other previous work on long-term trends of PM, particularly for rural areas.

However, to address reviewers' comments, we already added a statement emphasizing that the role of meteorology cannot be ruled out. We further included in this version the sentence "In most cases, there is a complex interplay between PM and meteorological conditions that further exacerbate PM mass concentration (Chen et al., 2020).", to strengthen this statement.

Regarding the added sentence, it is not clear what the authors mean by "further exacerbate PM mass concentration" and specifically how this is connected to the revealed (decreasing) trends.

2. As said, the novelty of the paper is the observation of the trends of the PM sources (apportioned using an enhanced PMF methodology) but also that of OP (apportioned using MLR). This allowed the unravelling of the decreasing trend in terms of source contributions by the STL model. The STL deconvolution was applied on all the identified sources, and it clearly shows that the traffic source has the highest tendency (see Table 1 in the manuscript) with a decreasing trend. The other major sources of PM10 (such as biomass burning, mineral dust, nitrate-, and sulphate-rich sources) do not have as much decreasing tendency as the traffic source. This is probably an indication that some of the processes included in "the internal annual variabilities of weather or climate conditions" are not leading factors in the temporal evolutions that can be seen (or not), since they would probably affect PM from different sources in the same way. Again, this aspect is not discussed further, since it is not our purpose to delineate these complex processes.

I would suggest to include part of the response in the revision.

3. However, thanks to the reviewers' comments, we added evidence in the second version that the decreasing traffic contributions is also in good agreement with the decreasing BC emissions from emissions inventory for France. Hence, we stand by one of our key take-aways stating that "While local or regional changes in meteorology may be a factor in the evolution of the concentrations observed, this

is unlikely to be the dominant one in the evolution of the concentrations of chemical species related to traffic emissions, in light of the strong correlation observed with the national emissions inventory in France.".

Ok.

**Anonymous Referee #1 (nominated 18 Mar 2022, accepted 25 Mar 2022, report 02 Apr 2022, Report #2):**

☐ Lines 34-35: This sentence is not clear: rephrase.

RESPONSE: The sentence in Line 34 to 35 is further improved into:

"Particulate matter (PM) pollution causes various environmental concerns affecting public health and climate."

Still not clear, in particular related with the climate effect of PM (since the dominant effect should be a cooling so compensating for global warming.

☐ Line 39: and what about aerosol transport apart from its formation?

RESPONSE: The sentence was improved and now reads as:

"Further work has also been carried out in more specific areas to understand particular processes of aerosol formation and transport, as well as specific sources such as in the boreal forest (Yan et al., 2016), polar environments (Barrie and Hoff, 1985; Moroni et al., 2016), high altitude (Rinaldi et al., 2015), or marine sites (Scerri et al., 2016)."

Ok.

☐ Line 45: well, it is quite restricting to say that the understanding of such processes is only related with the "elaboration" (perhaps also not the most appropriate term) of chemical transport model.

"Studies at such sites enable the understanding of large-scale and mesoscale processes (Anenberg et al., 2010; Mues et al., 2013; Konovalov et al., 2009), which is necessary to elaborate chemical transport models."

RESPONSE: The sentence was improved and now reads as:

"Studies at such sites provide more understanding of large-scale and mesoscale processes (Anenberg et al., 2010; Mues et al., 2013; Konovalov et al., 2009), which can be useful in the development and validation of chemical transport models."

Ok.

☐ Lines 47-48: This is a general characteristic of the long term time series, and not just of the ones collected at background sites.

RESPONSE: The phrase "in background sites" was removed.

Ok.

 Lines 71-73: This detail is not needed at this point.

"The PMF methodology includes a unique validation with comparison of the chemical profiles of the factors with those obtained in many other studies in France."

RESPONSE: This improvement (see sentence above) is a response to another referee's comment (Referee #2) that aims to emphasize the usefulness of this paper. In any case, we believe that it is good to mention that our work indeed included this amount of effort that enabled us to do a comparison of chemical profiles obtained in many different studies in France.

Ok.

This is a feature, allowing check for consistency, that is extremely rare in the literature. In fact, in Figure 5, we have included this discussion through the PD-SID metric.
Ok.

☐ Lines 69-77: The scopes are very specific and citing already methodologies which happen to be still rather obscure to the reader. Please try to generalize the objectives leaving the details of the methodology for later on in the text.

"The objectives of this work are, first, to achieve for the very first time a study of the main sources of PM in a rural environment in Europe, using a long-term database including several specific organic tracers in the carbonaceous fraction. The PMF methodology includes a unique validation with comparison of the chemical profiles of the factors with those obtained in many other studies in France. The second objective is to quantify the temporal evolution of the contributions of these sources over the period of the study, particularly focusing for the first time on the vehicular emission that have already been shown to decrease in urban environments in Europe during the last decades. Finally, another major objective is to perform the deconvolution of the contribution of the PM sources to the OP measured with AA and DDT assays, and to determine the most important sources for the oxidizing capabilities of PM influencing human health in such an environment."
RESPONSE: This paragraph has been improved and now reads as:
"The understanding of trends of PM sources are essential to evaluate the effects of mitigation policies on air pollution levels. A reference background site offers a good opportunity to gauge the broad effects of certain improvements in the transportation fleet and other regulations aimed at reducing vehicular emissions in large cities. Thus, in this study, an extensive dataset of PM over a 9-year period ($n = 434$), obtained from a French national background site, was investigated to: (1) provide insights on the long-term trends of PM sources and other emerging health-based metrics of PM exposure, such as OP of PM, (2) quantify the temporal evolution of the contributions of these sources, particularly focusing on vehicular emissions that have already been shown to decrease in urban environments in Europe during the last decades."
Ok.

☐ Line 140-141: Absolutely not clear how you reconstructed PM10.

RESPONSE: This was discussed in section 3.1, specifically mentioning that the reconstructed mass of PM10 in the OPE site was calculated following Eq. S1. In the SI are the equations used to reconstruct PM mass.
You can then provide a reference to the Supplementary Material for such description.

☐ Lines 142-144: This detail is not useful here, as the reader does not know anything of this comparison.

"A total of 299 out of 434 (69%) TEOM measurements were paired with reconstructed PM10 data, due to many interruptions in the TEOM functioning, in order to evaluate the semi volatile mass missing in the mass reconstruction with filter chemistry."

RESPONSE: This sentence (see sentence above) was added as an improvement following the comment of Referee #1, which suggested that the unknown portion of PM be added in Figure 2. In this sentence, the authors wanted to elaborate on how this was done.

Ok.

☐ Lines 206-207: There are other reasons to increase the uncertainties of some variables (indicating them as weak) or excluding them from the analysis; I assume that this has been considered. also, have other sources of uncertainty (e.g., flow rate) taken into account?

RESPONSE: Section 2.4.2 presented our criteria for a valid solution, which also mentioned that we followed the recommendations of the European guide on air pollution source apportionment with receptor models (Belis, 2019). This was the guideline that was followed, it presents in detail how to perform receptor modelling.

Each sample included was initially checked for its consistent and valid flow rate. We did not include samples that have questionable flow rates during sampling.

The guide does not have a specific value for the added extra uncertainty (to the whole dataset) and in any case it provides general guidelines on how the uncertainty can be determined but many details (e.g., evaluation of the S/N ratio, calculation of the uncertainty for missing data and for data below detection limits, etc.) are absolutely not fixed. I would suggest to include more details on this on the revised version of the manuscript.

☐ Line 211: How did you take into account the weighted residuals distribution? I mean, could you explain how you analysed it, such as you did with the Q/Qexp ratio?

RESPONSE: This information can be found in the PMF user guide 5.0 and the European guide on air pollution source apportionment with receptor models (Belis, 2019; mentioned in section 2.4.2 in the manuscript). In brief, the residual analysis is based on the uncertainty-scaled residuals. A specie is considered well-modelled, when all residuals are between +3 and -3 and are normally distributed. Species with residuals beyond +3 and -3 was evaluated in terms of their observed vs. predicted scatterplot and time-series analysis.

Ok, but perhaps a detail could be added in the revised version of the manuscript.

☐ Lines 265-284: Which software or code did you use to apply this analysis?

RESPONSE: It has been mentioned in the manuscript (section 2.6) that it was implemented in Python using the *statsmodels* module.

Ok.

◻ Lines 266-268: I cannot understand this: meteorology does not have only a seasonal signal, and also how can the interannual variation in the seasonal signal be connected with the effect of meteorology? If you have references, please provide them, because this justification is not convincing.

RESPONSE: Please refer to the general comments.
The response in the general comments does not address my specific comment above.

◻ Line 298: To be true, I can observe also a reduction in NO3- and NH4, which should be among the main chemical species. So I still cannot understand this.

RESPONSE: $NO_3^-$ and $NH_4^+$ are in fact among the main chemical species in Figure 2. Our point in this sentence was that the changes are not drastically changing through the years. In this paragraph, we were also discussing the yearly average volatile mass (i.e., unaccounted by chemical analysis).
I still see this as confusing, and I would suggest to make the sentence clearer in this regard.

◻ Lines 299-301: Can you explain at least tentatively the reasons of such differences?

RESPONSE: Our unaccounted portion is well within range generally found in a rural environment and we have added a reference that supports this. As discussed in many papers in the literature, the differences are generally attributed to all semi-volatile chemical species included in the PM (water vapor, organics, ammonium nitrate, …).
Detail on this should be added.

◻ Line 378: A part of sulphates has a marine origin.

"The **aged sea salt** factor is characterised by high loadings of $Na_+$ and $Mg_{2+}$, with a certain amount of species originating from potentially anthropogenic sources such as nitrates (6% of $NO_3^-$ mass) and sulphates (19% of $SO_{42-}$ mass) that can be attributed to mixing and transformation processes in the atmosphere."
RESPONSE: There is no statement in the manuscript that argues against "sulphates has marine origin". Indeed, the reviewer can observe that there is a fraction of sulfate included in the fresh sea -salt chemical profile (Figure S5), that largely increases in aged sea-salt (Figure S6).

◻ Lines 384-386: And what about the sulphates to $Na_+$ ratio? Did you observe if there is a particular wind direction for this factor? Or reasons to suspect collinearity? Or any other investigation on this factor which could be also a mixed source?

RESPONSE: We made it clear that we did not analyse meteorological data. Please refer to the general comments. Further, it is really difficult with the PMF (and nearly never discussed in papers with PMF results) to distinguish if the chemical profile of a factor includes some species (that are not fully known to be associated in a given source) because of co-linearity of sources or because the species are indeed

internally mixed in the PM because of interactions / modification during transport. In our case, with a multi-year data base, it seems unlikely that the presence of some fraction of OC in the MSA rich factor is present just because of collinearity or mix with another source, that would need to be maintained for the overall period.

The response is incomplete since the absence of meteorological data does not justify the absence of the analysis of the sulphates (or other species) to Na+ ratio.

☐ Lines 388-390: And what about the ratio of Cl- to Na+? Is there chlorine depletion?

REPONSE: Elaborating about the chlorine depletion does not add any useful information in terms of identifying factors resolved by PMF. However, the reviewer can observe that the chlorine that is present in the fresh sea-salt chemical profile (Figure S5) but is not apparent in the aged sea-salt profile (Figure S6), indicating chlorine depletion with ageing of the sea salt emissions.

Ok.

☐ Lines 413-416: And what about your study?

RESPONSE: In this paragraph, we talked about the description of primary biogenic factor. We talked about the characteristics of primary biogenic sources as reported in Samake et al. (2019), a paper also published by our group. This is a strong paper supporting the characteristics we have equally found in our study in the OPE site.

Ok.

☐ Lines 417-424: Any particular temporal pattern for this factor?

RESPONSE: Yes, it has temporal pattern and this is shown in Figure S1. In section 3.3, we have also mentioned that it is a major contributor during winter season.

☐ Line 432-434: Also here, any particular temporal pattern?

RESPONSE: There is a mild temporal pattern, please refer to Figure S4.

Ok.

☐ Lines 458-530: Apart from the analysis of trends, could you explain more how to interpret the results of the STL analysis for example in terms of different importance of the three signal components?

RESPONSE: Thank you for the suggestion, we have improved a part of section 2.6 that now reads as: "The STL (Season-trend deconvolution using locally estimated scatterplot smoothing) model is a versatile and robust statistical method allowing the decomposition of a time-series dataset into three components including trend, seasonality, and residual. The trend provides a general direction of the over-all data; the seasonality is a repeating pattern that recur over a fixed period of time; finally, residual is the random fluctuation or unpredictable change in the dataset."

This addition does not completely address my comment, since the information provided are generic for the STL analysis and does not refer explicitly to the results obtained here using this methodology.

☐ Figure 6: The Figure has poor resolution. Also, there is no unit of measurement on the y-axis.

**Figure 1: The Season-trend (STL) deconvolution of contributions in µg m$_{-3}$ from the traffic factor to PM$_{10}$ from year 2012 to 2020.**
RESPONSE: It is unclear if referee meant poor image resolution (which will be improved in the final paper) or he meant the resolution of the x-axis interval (which is essentially monthly resolution, fit for the purpose of the trend evaluation). The unit is given in the figure caption.
I was referring to the image resolution and to units of the y-axis.

☐ Line 579: Missing reference to a Figure.

RESPONSE: This has been corrected.
☐ Lines 660-666: This sentence is still not clear, and has to be revised.

RESPONSE: The paragraph has been improved and now reads as:
"There was a change in sampling duration between the collection performed in year 2012 to 2016 (7-day sampling) and 2016 to 2020 (24-hour sampling). A 7-day filter sample includes both weekdays and weekends, whereas a 24-hour sample will either be a weekday or weekend, depending on the sampling interval. This implies that the weekly collected samples may contain features that are not fully captured in a daily sample. However, since the OPE site is quite distant from direct emissions, the expected difference in the weekday and weekend levels should be relatively small. Further, PMF source apportionment were conducted separately on the two periods (i.e., 7-day samples versus 24-hour samples), leading to very similar results for the chemical profiles and source contributions, justifying the coupled analysis."
Ok.

☐ Lines 702-703: Do you mean the improvements in the technology?

RESPONSE: Yes, technology in various aspects.

☐ Lines 704-705: Not clear what you mean by "persistent changes". Revise.

RESPONSE: This sentence was improved and now reads as:

"However, persistent changes in meteorological conditions influencing the transport of air masses to OPE or formation of PM during this transport cannot be totally ruled out."

☐ Code and data availability: still not in line with the policy of this journal.

RESPONSE: Our declaration is that these "could be made available upon request by contacting the corresponding author" in order to be in line with university and research institution policies and legal terms with our funding groups.

☐ Table S2 and text: the "nitrate-rich" and "sulfate-rich" factors should be "secondary nitrate" and "secondary sulfates"? also, the "fresh sea salt" and "aged sea salt" share much of the fingerprint, so there is no clear evidence of why they should be separated in two factors, at least from this table.

RESPONSE: The authors deem that the two factors, nitrate-rich and sulphate-rich, are named accurately (as discussed in the description of factors in section 3.3), there is no added value in changing the names as suggested by the referee. Those are terms widely accepted and recognized by all research groups doing PMF studies. Sulfate- and nitrate-rich names were coined because these factors do not only contain sulfate or nitrate secondary components but also aggregate some other chemicals species. As for fresh and aged sea salt, one of the major differences is the depletion of chlorine (as mentioned above), but also the aggregation of other components (like sulfate, also mentioned above).

☐ Figure S1 and text: is the temporal variation of biomass burning in agreement with your expectations? Does it increase in winter (it seems so) and with wildfires?

RESPONSE: We already mentioned in the main text that the Biomass burning factor is a major contributor during winter. There are not many reported wildfires in proximity to the sampling site, if any it would be relatively more common in South of France (~700 km away). Agricultural fires (which are not so common in France) could be possible to explain some low level contribution out of the winter season, where the emissions are totally dominated by domestic heating with wood burning.

☐ Figure S4: About the mineral dust factor (possibly better named "resuspension", as mineral dust would be just the transport of dust from the desert), I would expect an increase in the summer season with less precipitation, and with Saharan dust transports. Is this the case?

RESPONSE: It was also mentioned in the main text that there could be an influence from re-suspended road dust. Yes, there is somehow a seasonal pattern, but not as clear as the pattern seen in the Biomass burning factor. We can assume the influence of precipitation and Saharan

dust transport; this has been extensively published in the past already. The authors deem it is unnecessary to provide meteorological data just so we can support this phenomenon.

☐ Figure S1-S9: In most cases the BS and DISP bars (in the legend, but are actually lines?) cannot be seen. Also, apart from the temporal pattern, could you investigate the effect of meteorology at least in some of these factors?

RESPONSE: The BS and DISP bars are very close to the contributions estimated in the final constrained PMF solution (100% mapping, discussed in section 3.2). Please refer to the general comments for our response about meteorological data.

---

## Author Response (AR2)

**9-year trends of PM$_{10}$ sources and oxidative potential in a rural background site in France**

Authors' response

We would like to thank the referees for their time to re-evaluate our manuscript and the improvements made during the first round of review. Our point-by-point response to the second round of comments are presented below (in blue).

We have also summarized a general comment with these specific points/clarifications:

1. Our reviewers have been insisting on trying to delineate the "trends that were caused by internal annual variabilities of weather or climate conditions". However, it was assumed already in the initial version that the search for all the causes of the temporal evolutions of source contributions was not in the scope of our work, and it is still not in the direction that we want for this paper. This would be a totally different work, that would require gathering many other data (on local and large-scale meteorology, on emission inventories, on parameters that are influencing atmospheric chemistry for the formation of secondary chemical species, …) on the long term, and it is not our purpose. One aspect of our paper is focused on showing that the data set allows to identify trends in the evolution of the sources contributions of PM for this rural site, and that these trends are different for one source to another. All of this is already an innovative step compared to most of the (few) similar papers in the literature that discuss of evolutions of the concentrations of some chemical species for similar sites, or evolutions of sources with a set of tracers that is much less elaborated than ours. We believe that this innovative result by itself justify the publication, as a step ahead of other previous work on long-term trends of PM, particularly for rural areas.

However, to address reviewers' comments, we already added a statement emphasizing that the role of meteorology cannot be ruled out. We further included in this version the sentence "In most cases, there is a complex interplay between PM and meteorological conditions that further exacerbate PM mass concentration (Chen et al., 2020).", to strengthen this statement.

2. As said, the novelty of the paper is the observation of the trends of the PM sources (apportioned using an enhanced PMF methodology) but also that of OP (apportioned using MLR). This allowed the unravelling of the decreasing trend in terms of source contributions by the STL model. The STL deconvolution was applied on all the identified sources, and it clearly shows that the traffic source has the highest tendency (see Table 1 in the manuscript) with a decreasing trend. The other major sources of PM10 (such as biomass burning, mineral dust, nitrate-, and sulphate-rich sources) do not have as much decreasing tendency as the traffic source. This is probably an indication that some of the processes included in "the internal annual variabilities of weather or climate conditions" are not leading factors in the temporal evolutions that can be seen (or not), since they would probably affect PM from different sources in the same way. Again, this aspect is not discussed further, since it is not our purpose to delineate these complex processes.

3. However, thanks to the reviewers' comments, we added evidence in the second version that the decreasing traffic contributions is also in good agreement with the decreasing BC emissions from emissions inventory for France. Hence, we stand by one of our key take-aways stating that "While local or regional changes in meteorology may be a factor in the evolution of the concentrations observed, this is unlikely to be the dominant one in the evolution of the

concentrations of chemical species related to traffic emissions, in light of the strong correlation observed with the national emissions inventory in France.".

**Anonymous Referee #1 (nominated 18 Mar 2022, accepted 25 Mar 2022, report 02 Apr 2022, Report #2):**

- Lines 34-35: This sentence is not clear: rephrase.

RESPONSE: The sentence in Line 34 to 35 is further improved into:

"Particulate matter (PM) pollution causes various environmental concerns affecting public health and climate."

- Line 39: and what about aerosol transport apart from its formation?

RESPONSE: The sentence was improved and now reads as:

"Further work has also been carried out in more specific areas to understand particular processes of aerosol formation and transport, as well as specific sources such as in the boreal forest (Yan et al., 2016), polar environments (Barrie and Hoff, 1985; Moroni et al., 2016), high altitude (Rinaldi et al., 2015), or marine sites (Scerri et al., 2016)."

- Line 45: well, it is quite restricting to say that the understanding of such processes is only related with the "elaboration" (perhaps also not the most appropriate term) of chemical transport model.

"Studies at such sites enable the understanding of large-scale and mesoscale processes (Anenberg et al., 2010; Mues et al., 2013; Konovalov et al., 2009), which is necessary to elaborate chemical transport models."

RESPONSE: The sentence was improved and now reads as:

"Studies at such sites provide more understanding of large-scale and mesoscale processes (Anenberg et al., 2010; Mues et al., 2013; Konovalov et al., 2009), which can be useful in the development and validation of chemical transport models."

- Lines 47-48: This is a general characteristic of the long term time series, and not just of the ones collected at background sites.

RESPONSE: The phrase "in background sites" was removed.

- Lines 71-73: This detail is not needed at this point.

"The PMF methodology includes a unique validation with comparison of the chemical profiles of the factors with those obtained in many other studies in France."

RESPONSE: This improvement (see sentence above) is a response to another referee's comment (Referee #2) that aims to emphasize the usefulness of this paper. In any case, we believe that it is good to mention that our work indeed included this amount of effort that enabled us to do a comparison of chemical profiles obtained in many different studies in France.

This is a feature, allowing check for consistency, that is extremely rare in the literature. In fact, in Figure 5, we have included this discussion through the PD-SID metric.

- Lines 69-77: The scopes are very specific and citing already methodologies which happen to be still rather obscure to the reader. Please try to generalize the objectives leaving the details of the methodology for later on in the text.

"The objectives of this work are, first, to achieve for the very first time a study of the main sources of PM in a rural environment in Europe, using a long-term database including several specific organic tracers in the carbonaceous fraction. The PMF methodology includes a unique validation with comparison of the chemical profiles of the factors with those obtained in many other studies in France. The second objective is to quantify the temporal evolution of the contributions of these sources over the period of the study, particularly focusing for the first time on the vehicular emission that have already been shown to decrease in urban environments in Europe during the last decades. Finally, another major objective is to perform the deconvolution of the contribution of the PM sources to the OP measured with AA and DDT assays, and to determine the most important sources for the oxidizing capabilities of PM influencing human health in such an environment."

RESPONSE: This paragraph has been improved and now reads as:

"The understanding of trends of PM sources are essential to evaluate the effects of mitigation policies on air pollution levels. A reference background site offers a good opportunity to gauge the broad effects of certain improvements in the transportation fleet and other regulations aimed at reducing vehicular emissions in large cities. Thus, in this study, an extensive dataset of PM over a 9-year period ($n = 434$), obtained from a French national background site, was investigated to: (1) provide insights on the long-term trends of PM sources and other emerging health-based metrics of PM exposure, such as OP of PM, (2) quantify the temporal evolution of the contributions of these sources, particularly focusing on vehicular emissions that have already been shown to decrease in urban environments in Europe during the last decades."

- Line 140-141: Absolutely not clear how you reconstructed PM10.

RESPONSE: This was discussed in section 3.1, specifically mentioning that the reconstructed mass of PM10 in the OPE site was calculated following Eq. S1. In the SI are the equations used to reconstruct PM mass.

- Lines 142-144: This detail is not useful here, as the reader does not know anything of this comparison.

"A total of 299 out of 434 (69%) TEOM measurements were paired with reconstructed PM10 data, due to many interruptions in the TEOM functioning, in order to evaluate the semi volatile mass missing in the mass reconstruction with filter chemistry."

RESPONSE: This sentence (see sentence above) was added as an improvement following the comment of Referee #1, which suggested that the unknown portion of PM be added in Figure 2. In this sentence, the authors wanted to elaborate on how this was done.

- Lines 206-207: There are other reasons to increase the uncertainties of some variables (indicating them as weak) or excluding them from the analysis; I assume that this has

been considered. also, have other sources of uncertainty (e.g., flow rate) taken into account?

RESPONSE: Section 2.4.2 presented our criteria for a valid solution, which also mentioned that we followed the recommendations of the European guide on air pollution source apportionment with receptor models (Belis, 2019). This was the guideline that was followed, it presents in detail how to perform receptor modelling.

Each sample included was initially checked for its consistent and valid flow rate. We did not include samples that have questionable flow rates during sampling.

- Line 211: How did you take into account the weighted residuals distribution? I mean, could you explain how you analysed it, such as you did with the Q/Qexp ratio?

RESPONSE: This information can be found in the PMF user guide 5.0 and the European guide on air pollution source apportionment with receptor models (Belis, 2019; mentioned in section 2.4.2 in the manuscript). In brief, the residual analysis is based on the uncertainty-scaled residuals. A specie is considered well-modelled, when all residuals are between +3 and -3 and are normally distributed. Species with residuals beyond +3 and -3 was evaluated in terms of their observed vs. predicted scatterplot and time-series analysis.

- Lines 265-284: Which software or code did you use to apply this analysis?

RESPONSE: It has been mentioned in the manuscript (section 2.6) that it was implemented in Python using the *statsmodels* module.

- Lines 266-268: I cannot understand this: meteorology does not have only a seasonal signal, and also how can the interannual variation in the seasonal signal be connected with the effect of meteorology? If you have references, please provide them, because this justification is not convincing.

RESPONSE: Please refer to the general comments.

- Line 298: To be true, I can observe also a reduction in NO3- and NH4, which should be among the main chemical species. So I still cannot understand this.

RESPONSE: $NO_3^-$ and $NH_4^+$ are in fact among the main chemical species in Figure 2. Our point in this sentence was that the changes are not drastically changing through the years. In this paragraph, we were also discussing the yearly average volatile mass (i.e., unaccounted by chemical analysis).

- Lines 299-301: Can you explain at least tentatively the reasons of such differences?

RESPONSE: Our unaccounted portion is well within range generally found in a rural environment and we have added a reference that supports this. As discussed in many papers in the literature, the differences are generally attributed to all semi-volatile chemical species included in the PM (water vapor, organics, ammonium nitrate, …).

- Line 378: A part of sulphates has a marine origin.

"The **aged sea salt** factor is characterised by high loadings of $Na^+$ and $Mg^{2+}$, with a certain amount of species originating from potentially anthropogenic sources such as nitrates (6% of

NO$_3^-$ mass) and sulphates (19% of SO$_4^{2-}$ mass) that can be attributed to mixing and transformation processes in the atmosphere."

RESPONSE: There is no statement in the manuscript that argues against "sulphates has marine origin". Indeed, the reviewer can observe that there is a fraction of sulfate included in the fresh sea -salt chemical profile (Figure S5), that largely increases in aged sea-salt (Figure S6).

- Lines 384-386: And what about the sulphates to Na+ ratio? Did you observe if there is a particular wind direction for this factor? Or reasons to suspect collinearity? Or any other investigation on this factor which could be also a mixed source?

RESPONSE: We made it clear that we did not analyse meteorological data. Please refer to the general comments. Further, it is really difficult with the PMF (and nearly never discussed in papers with PMF results) to distinguish if the chemical profile of a factor includes some species (that are not fully known to be associated in a given source) because of co-linearity of sources or because the species are indeed internally mixed in the PM because of interactions / modification during transport. In our case, with a multi-year data base, it seems unlikely that the presence of some fraction of OC in the MSA rich factor is present just because of collinearity or mix with another source, that would need to be maintained for the overall period.

- Lines 388-390: And what about the ratio of Cl- to Na+? Is there chlorine depletion?

REPONSE: Elaborating about the chlorine depletion does not add any useful information in terms of identifying factors resolved by PMF. However, the reviewer can observe that the chlorine that is present in the fresh sea-salt chemical profile (Figure S5) but is not apparent in the aged sea-salt profile (Figure S6), indicating chlorine depletion with ageing of the sea salt emissions.

- Lines 413-416: And what about your study?

RESPONSE: In this paragraph, we talked about the description of primary biogenic factor. We talked about the characteristics of primary biogenic sources as reported in Samake et al. (2019), a paper also published by our group. This is a strong paper supporting the characteristics we have equally found in our study in the OPE site.

- Lines 417-424: Any particular temporal pattern for this factor?

RESPONSE: Yes, it has temporal pattern and this is shown in Figure S1. In section 3.3, we have also mentioned that it is a major contributor during winter season.

- Line 432-434: Also here, any particular temporal pattern?

RESPONSE: There is a mild temporal pattern, please refer to Figure S4.

- Lines 458-530: Apart from the analysis of trends, could you explain more how to interpret the results of the STL analysis for example in terms of different importance of the three signal components?

RESPONSE: Thank you for the suggestion, we have improved a part of section 2.6 that now reads as:

"The STL (Season-trend deconvolution using locally estimated scatterplot smoothing) model is a versatile and robust statistical method allowing the decomposition of a time-series dataset

into three components including trend, seasonality, and residual. The trend provides a general direction of the over-all data; the seasonality is a repeating pattern that recur over a fixed period of time; finally, residual is the random fluctuation or unpredictable change in the dataset."

- Figure 6: The Figure has poor resolution. Also, there is no unit of measurement on the y-axis.

[Figure]

**Figure 1: The Season-trend (STL) deconvolution of contributions in μg m⁻³ from the traffic factor to PM₁₀ from year 2012 to 2020.**

RESPONSE: It is unclear if referee meant poor image resolution (which will be improved in the final paper) or he meant the resolution of the x-axis interval (which is essentially monthly resolution, fit for the purpose of the trend evaluation). The unit is given in the figure caption.

- Line 579: Missing reference to a Figure.

RESPONSE: This has been corrected.

- Lines 660-666: This sentence is still not clear, and has to be revised.

RESPONSE: The paragraph has been improved and now reads as:

"There was a change in sampling duration between the collection performed in year 2012 to 2016 (7-day sampling) and 2016 to 2020 (24-hour sampling). A 7-day filter sample includes both weekdays and weekends, whereas a 24-hour sample will either be a weekday or weekend, depending on the sampling interval. This implies that the weekly collected samples may contain features that are not fully captured in a daily sample. However, since the OPE site is quite distant from direct emissions, the expected difference in the weekday and weekend levels should be relatively small. Further, PMF source apportionment were conducted separately on the two periods (i.e., 7-day samples versus 24-hour samples), leading to very similar results for the chemical profiles and source contributions, justifying the coupled analysis."

- Lines 702-703: Do you mean the improvements in the technology?

RESPONSE: Yes, technology in various aspects.

- Lines 704-705: Not clear what you mean by "persistent changes". Revise.

RESPONSE: This sentence was improved and now reads as:

"However, persistent changes in meteorological conditions influencing the transport of air masses to OPE or formation of PM during this transport cannot be totally ruled out."

- Code and data availability: still not in line with the policy of this journal.

RESPONSE: Our declaration is that these "could be made available upon request by contacting the corresponding author" in order to be in line with university and research institution policies and legal terms with our funding groups.

- Table S2 and text: the "nitrate-rich" and "sulfate-rich" factors should be "secondary nitrate" and "secondary sulfates"? also, the "fresh sea salt" and "aged sea salt" share much of the fingerprint, so there is no clear evidence of why they should be separated in two factors, at least from this table.

RESPONSE: The authors deem that the two factors, nitrate-rich and sulphate-rich, are named accurately (as discussed in the description of factors in section 3.3), there is no added value in changing the names as suggested by the referee. Those are terms widely accepted and recognized by all research groups doing PMF studies. Sulfate- and nitrate-rich names were coined because these factors do not only contain sulfate or nitrate secondary components but also aggregate some other chemicals species. As for fresh and aged sea salt, one of the major differences is the depletion of chlorine (as mentioned above), but also the aggregation of other components (like sulfate, also mentioned above).

- Figure S1 and text: is the temporal variation of biomass burning in agreement with your expectations? Does it increase in winter (it seems so) and with wildfires?

RESPONSE: We already mentioned in the main text that the Biomass burning factor is a major contributor during winter. There are not many reported wildfires in proximity to the sampling site, if any it would be relatively more common in South of France (~700 km away). Agricultural fires (which are not so common in France) could be possible to explain some low level contribution out of the winter season, where the emissions are totally dominated by domestic heating with wood burning.

- Figure S4: About the mineral dust factor (possibly better named "resuspension", as mineral dust would be just the transport of dust from the desert), I would expect an increase in the summer season with less precipitation, and with Saharan dust transports. Is this the case?

RESPONSE: It was also mentioned in the main text that there could be an influence from re-suspended road dust. Yes, there is somehow a seasonal pattern, but not as clear as the pattern seen in the Biomass burning factor. We can assume the influence of precipitation and Saharan

dust transport; this has been extensively published in the past already. The authors deem it is unnecessary to provide meteorological data just so we can support this phenomenon.

- Figure S1-S9: In most cases the BS and DISP bars (in the legend, but are actually lines?) cannot be seen. Also, apart from the temporal pattern, could you investigate the effect of meteorology at least in some of these factors?

RESPONSE: The BS and DISP bars are very close to the contributions estimated in the final constrained PMF solution (100% mapping, discussed in section 3.2). Please refer to the general comments for our response about meteorological data.

**Anonymous Referee #2 (nominated 18 Mar 2022, accepted 21 Mar 2022, report 31 Mar 2022, Report #1):**

Overall, I believe the authors have made an effort to improve the quality of the manuscript and they have addressed most of my concerns. I still cannot say that I have learnt much from reading the revised version of the manuscript, but I do acknowledge that there is some value in analyzing 9 years of data of chemical speciation and 4 years of OP, although the final results are not surprising. I still have a few minor comments/suggestions left, which I am summarizing below:

One thing that I recommend is being more careful and specific when crafting the research questions and scientific objectives of the manuscript, even for future work. The purpose of a good scientific paper should not be to perform the analysis just for its own sake, but the analysis should serve to answer a question or prove an hypothesis. Sentences like the following ones, in your revised manuscript:

'Finally, another major objective is to perform the deconvolution of the contribution of the PM sources to the OP measured with AA and DDT assays …'

or

'The objectives of this work are, first, to achieve for the very first time a study of the main sources of PM in a low altitude rural environment in Europe, using a long-term database including several specific organic tracers in the carbonaceous fraction…'

Are related to the methods, the analysis and the dataset itself, but say nothing about the scientific problem that you're trying to solve/address.

Your introduction should be probably something along the lines of:

'As a community, we do not know how the chemical speciation of PM10 changed in recent years in rural sites, and it would be interesting to gauge that because of reasons x,y,z. Thus, we analyze a novel dataset to answer the following questions: (1) … (2) … '. The objective of a scientific paper cannot be performing an analysis, but it should be to uncover something unknown through some analysis.

RESPONSE: We appreciate the suggestion of the reviewer and have improved part of section 1 as follows:

"The understanding of trends of sources of PM are essential to evaluate the effects of mitigation policies on air pollution levels. Particularly, a reference background site offers a good

opportunity to gauge the broad effects of certain improvements in the transportation fleet and other regulations aimed at reducing vehicular emissions in large cities. Thus, in this study, an extensive dataset of PM over a 9-year period ($n$ = 434), obtained from a French national background site, was investigated to: (1) provide insights on the long-term trends of PM sources and other emerging health-based metrics of PM exposure, such as OP of PM, (2) quantify the temporal evolution of the contributions of these sources, particularly focusing on vehicular emissions that have already been shown to decrease in urban environments in Europe during the last decades."

About the new Figure 7 in the manuscript:

I was glad to see an interesting agreement between BC emissions and traffic-related PM10 trends. However, I believe that the presentation quality of Figure 7 can be enhanced (lack of y-axis, low resolution, unclear legend (is it BC emissions?, x-axis, tick-marks and so on).

Although the correlation is striking, I'm still a bit puzzled by the comparison between an indirect quantity (traffic-related PM10 is derived from PMF) and BC emissions. Perhaps a more straightforward analysis, which would strengthen the validity of your argument, would be comparing the trends of EC emissions in France and EC concentrations measured in your samples throughout the nine-year period. This would be helpful to see because EC concentrations are not derived from PMF and thus represent a more direct quantity to assess the impact of vehicular emissions.

RESPONSE:

It is true that one perfectly relevant comparison would be between BC from vehicular emissions from an inventory on one side versus the measured fraction of EC coming from the vehicular emissions on the other side. However, we do not have such measurement of EC from vehicular emissions, since the EC measured can come from all combustion sources, and also from dust resuspension. It is the heart of the paper to delineate the sources, and we therefore believe that the comparison of the trend of our vehicular PM10 factor is the most appropriate with a proxy of the estimate emissions. And we agree with the reviewer that the comparison is striking.

3. Lines 540-542: There are some missing references there denoted as 'Erreur ! Source du renvoi introuvable'

RESPONSE: This has been corrected in the manuscript.

---

## Author Response (AR3)

**9-year trends of PM$_{10}$ sources and oxidative potential in a rural background site in France**

**Authors' response**

We would like to thank the referees for their time to re-evaluate our manuscript and the improvements made during the first and second rounds of review. Our point-by-point response to the third round of comments are presented below (in blue).

**Anonymous Referee #2 nominated 25 May 2022, accepted 30 May 2022, report 30 May 2022 (Report #1):**

- Thanks for addressing my concerns, especially the introduction-related ones. The research questions and goals of the paper are outlined much better now, and I do think that the authors are addressing an interesting question. The only suggestion that I have is to improve the quality of Figure 7, as I noted in the previous round of review (tick marks, y-axis is missing, low-resolution, etc.) -- Scientific papers should also have good quality in presenting the data.

RESPONSE: We appreciate the feedback of the referee. We have improved Figure 7 as:

[Figure]

**Anonymous Referee #1 nominated 25 May 2022, accepted 01 Jun 2022, report 06 Jun 2022 (Report #2):**

- Lines 431: Regarding the added sentence, it is not clear what the authors mean by "further exacerbate PM mass concentration" and specifically how this is connected to the revealed (decreasing) trends.

RESPONSE: Line 429 to 436 is a paragraph discussing the interplay between meteorological conditions and PM. To make it clearer, the specific sentence mentioned above was improved and now the paragraph reads as:

It should be noted that the role of meteorology on the observed decrease in PM in these studies (including ours) cannot be totally ruled out (Hou and Wu, 2016; Czernecki et al., 2017; Kim, 2019) and is generally not fully considered. In most cases, there is a complex interplay between

PM and meteorological conditions that could increase or decrease PM mass concentration (Chen et al., 2020). Indeed, there are some studies at high-altitude or regional background sites that highlighted a concurrent role of changing large scale meteorology and changes in frequency of Saharan dust advections to Europe (Brattich et al., 2020) in modulating the dust concentrations in the atmosphere. The study at Melpitz (Spindler et al., 2013), despite an in-depth work on the wind sector classification, does not address the impact of possible changing in the air mass origin on long-term changing concentrations.

- General comment #2 from second round of review: I would suggest to include part of the response in the revision.

RESPONSE: Accordingly, a paragraph was added in section 3.5 that reads as:

"These findings allowed the unravelling of the decreasing trend in terms of source contributions by the STL model. The STL deconvolution was applied on all the identified sources, which clearly showed that the traffic source has the highest tendency with a decreasing trend. The other major sources of PM, such as biomass burning, mineral dust, nitrate-rich sources, do not have as much decreasing tendency as the traffic factor. The internal annual variabilities of weather/climate conditions might not be the leading factors explaining these trends, as they would have affected PM sources in the same way."

- Line 34 to 35: "Particulate matter (PM) pollution causes various environmental concerns affecting public health and climate.": Still not clear, in particular related with the climate effect of PM (since the dominant effect should be a cooling so compensating for global warming.

RESPONSE: We understand the concern of the reviewer. PM is composed of a wide range of species that can have either warming or cooling effects on the climate, nevertheless air pollution and climate influence each other through complex interactions in the atmosphere—which the authors deem unnecessary to elaborate on the first sentence of the introduction.

- You can then provide a reference to the Supplementary Material for such description.

RESPONSE: Please refer to Line 237 that reads: "The reconstructed mass of $PM_{10}$ in the OPE site was calculated following Eq. S1 in the SI and is presented in Figure 2."

- Section 2.4.2: The guide does not have a specific value for the added extra uncertainty (to the whole dataset) and in any case it provides general guidelines on how the uncertainty can be determined but many details (e.g., evaluation of the S/N ratio, calculation of the uncertainty for missing data and for data below detection limits, etc.) are absolutely not fixed. I would suggest to include more details on this on the revised version of the manuscript.

RESPONSE: We appreciate the feedback of the reviewer. We have further improved section S3 in the supplementary information which now includes more information about the PMF methodology, that reads as:

"For some species, it was necessary to use an expanded uncertainty that takes into account analytical error and sampling error, which can be used instead of the methodology proposed by (Gianini et al., 2012). An uncertainty of $\frac{5}{6} \times DL$ was used for values <DL and the uncertainties that are four times the specie concentration geometric mean were attributed to missing or replaced values.

The robustness of the final PMF solution was evaluated using various statistical parameters based on the European guide on air pollution source apportionment with receptor models (Belis et al., 2014) and the geochemical soundness of the solution. In brief, the parameters are listed as follows:

- ✓ Evolution of the ratio Qtrue/Qrobust (<1.5)

- ✓ The weighted residuals are normally distributed for most of the species and between ±3 which should indicate good model results of most variables

- ✓ Evaluation of the statistical robustness of the optimal solution (sensitivity to noise and any random data point) using a bootstrap test (BS) for 100 successive iterations of the model and for a minimum correlation ($r^2$) of 0.6

- ✓ Evaluation of the geochemical soundness of the PMF-resolved factor profiles based on *a priori* knowledge of the chemical footprints of the sources, their specific tracers, the temporal variability (daily, weekly and seasonally), and the characteristics of the site studied

- ✓ Statistical evaluation and precision for constrained solutions using BS for 100 successive iterations of the model and for a minimum correlation ($r^2$) of 0.6

- ✓ There is no added extra uncertainty to the whole dataset"

- • Ok, but perhaps a detail could be added in the revised version of the manuscript.

RESPONSE: Please refer to response above.

- • Lines 266-268: I cannot understand this: meteorology does not have only a seasonal signal, and also how can the interannual variation in the seasonal signal be connected with the effect of meteorology? If you have references, please provide them, because this justification is not convincing. The response in the general comments does not address my specific comment above.

RESPONSE: In the second round of review, we have already addressed that the search for all the causes of the trends by internal annual variabilities of weather or climate conditions was not in the scope of our work, and it is still not in the direction that we want for this paper.

- • Line 298: I still see this as confusing, and I would suggest to make the sentence clearer in this regard.

RESPONSE: Previously, the reviewer mentioned that $NO_3^-$ and $NH_4^+$ should be among the main chemical species. Our response to that was to clarify that, indeed, these two species are

among the main chemical species (you can also refer to Figure 2 in the manuscript). For clarity, we have improved the sentence that now reads as:

"Some changes in the concentration can be observed in the PM10 mass concentration, but there are no drastic changes in the major chemical components at the OPE, even with the lockdown restrictions during year 2020".

- Detail on this should be added.

RESPONSE: The unaccounted portion of PM found in this study is well within range of other rural environments as supported by the reference that we have provided. The authors deem it unnecessary to elaborate further on fractions that are, as it is, unaccounted.

- Lines 384-386: And what about the sulphates to Na+ ratio? Did you observe if there is a particular wind direction for this factor? Or reasons to suspect collinearity? Or any other investigation on this factor which could be also a mixed source?: The response is incomplete since the absence of meteorological data does not justify the absence of the analysis of the sulphates (or other species) to Na+ ratio.

RESPONSE: We made it clear that we did not analyse meteorological data. This statement addresses the question, "Did you observe if there is a particular wind direction for this factor?". As discussed in Section 3.2, constraints were used in the final model (Table S3), which resulted to all factors being correctly mapped and all bootstrap runs converged, thereby showing overall statistical robustness of the model. The chemical profile and temporal evolution including the reference run, bootstrap and displacement error estimates of each identified factor were provided in the S3 in the supplementary information. The PMF solution description and factor contributions were also provided in Section 3.3. For example, the aged sea salt factor is characterised by high loadings of $Na^+$ and $Mg^{2+}$, with a certain amount of species originating from potentially anthropogenic sources such as nitrates (6% of $NO_3^-$ mass) and sulphates (19% of $SO_4^{2-}$ mass) that can be attributed to mixing and transformation processes in the atmosphere. Interestingly, there are some contributions from EC (8% of EC mass), Cu (11% of Cu mass), Sb (13% of Sb mass), and Se (19% of Se mass). This could imply potential mixing of aged sea salt with other anthropogenic source linked to these species (e.g., traffic, shipping). But even with the possible mixing with anthropogenic source, this factor is clearly aged sea salt based on its chemical profile and factor contributions (Figure S6).

Again, the authors would like to point out that it is really difficult to differentiate if species are indeed internally mixed in the PM because of interactions / modification during transport or if there are mixing issues in the PMF solution between sources. This is why it is important that the chemical profiles in the OPE site were compared with other existing chemical profiles in other sites in France as shown in Figure 5 (similarity plot (PD-SID metric) of the OPE site against all the French sites in the SOURCES programme). This is probably the best test on the robustness of the factor and it is very rarely considered in any PMF paper in the literature.

- Apart from the analysis of trends, could you explain more how to interpret the results of the STL analysis for example in terms of different importance of the three signal components? : This addition does not completely address my comment, since the information provided are generic for the STL analysis and does not refer explicitly to the results obtained here using this methodology. but perhaps a detail could be added in the revised version of the manuscript.

RESPONSE: We understand the reviewer's concern and have further improved section 2.6 which now reads as:

"The STL (Season-trend deconvolution using locally estimated scatterplot smoothing) model is a versatile and robust statistical method allowing the decomposition of a time-series dataset into three components including trend, seasonality, and residual. The trend provides a general direction of the over-all data; the seasonality is a repeating pattern that recur over a fixed period of time; finally, residual is the random fluctuation or unpredictable change in the dataset. The seasonal component allows to eliminate seasonal variation from the time series, resulting to a smoothed trend line that shows the tendency of the time-series dataset. This method somehow takes into account the changes in seasonal cycles from year to year which could also delineate part of the effect of meteorology on the long-term trend of $PM_{10}$."

- I was referring to the image resolution and to units of the y-axis.

RESPONSE: The units are provided in the figure caption. The image resolution was increased to 300 dpi.

[Figure]

STL deconvolution of the Traffic factor in the OPE site

$y=-0.009x + 1.608$ ($r^2=0.67$), so $-0.104$ nmol min$^{-1}$ m$^{-3}$ an$^{-1}$